# A Unified Framework for EEG–Video Emotion Recognition with Brain Anatomy Guidance

## Abstract

Recent studies in video- and EEG-based emotion recognition have shown notable progress. However, multi-modal emotion recognition remains largely unexplored, particularly the integration of physiological signals with video. This integration is crucial, as EEG–video fusion combines observable behavioral cues with internal neural dynamics and enables a more comprehensive and robust characterization of human emotion. To this end, we propose EVER, a novel EEG–Video Emotion Recognition framework that effectively integrates complementary information from both modalities. Specifically, EVER employs a Brain anatomy-aware Inter-modal Hierarchical Graph Convolution Network (BIH-GCN), which aggregates EEG channel features into region-level representations guided by anatomical priors. These region-level features are combined with global high-level EEG and video backbone embeddings to form a unified representation for emotion classification. Furthermore, we introduce a correlation-based distribution alignment loss to reconcile modality-specific embeddings and reduce cross-modal discrepancies. To provide a comprehensive evaluation, we conduct comprehensive benchmark across three public EEG-video paired datasets—Emognition, MDMER, and EAV. We evaluate 12 representative models, consisting of 5 EEG-only, 5 video-only, and 2 audio-video models, and report their performance under EEG, video, and EEG–video settings. Our benchmark highlights the strengths and limitations of both unimodal and multi-modal approaches across diverse environments. Extensive experiments demonstrate that the proposed EVER achieves state-of-the-art performance by jointly modeling behavioral cues from video and physiological responses from EEG, thereby enabling the recognition of emotional patterns unattainable by either modality alone.

## 1 Introduction

Emotion recognition is a fundamental task in numerous real-world applications, including mental health monitoring (Aina et al., 2024), human-computer interaction (Chowdary et al., 2023), and affective computing (Cortiñas-Lorenzo & Lacey, 2023). Recent advances leveraging video (Zhang et al., 2021; Ben et al., 2021) or EEG signals (Peng et al., 2022; Liu et al., 2024a) have demonstrated the potential of each modality to provide valuable insights into human emotion. Video captures observable behavioral cues such as facial expressions (Zhang et al., 2020), whereas EEG reflects neural dynamics associated with internal states (Lee et al., 2024). These modalities have typically been studied in isolation, and most existing datasets and benchmarks (Koelstra et al., 2011; Zheng & Lu, 2015; Zafeiriou et al., 2017; Katsigiannis & Ramzan, 2017) are designed for unimodal learning.

However, unimodal networks often struggle to generalize across various scenarios, and relying solely on either modality provides only a partial and fragile view of human emotion. For instance, video-based methods can be hindered when individuals intentionally mask their emotions (Iwasaki & Noguchi, 2016) or when visual cues are ambiguous due to occlusion or adverse lighting conditions (Elsheikh et al., 2024). Conversely, EEG-based methods provide an internal and less ambiguous perspective on affective states, yet they are limited by inter-subject variability and sensor-related artifacts (Peng et al., 2022). Although the value of multi-modal fusion has been widely recognized in related domains such as audio–video emotion recognition (Noroozi et al., 2017; Guanghui & Xiaoping, 2021; Tang et al., 2022; Sun et al., 2024; Wu et al., 2025), the integration of video with physiological signals like EEG remains largely unexplored due to scarce research efforts.

Therefore, we introduce a novel EEG–Video Emotion Recognition (EVER) framework that explicitly unifies video and EEG representations to overcome the limitations of existing multi-modal architectures across heterogeneous modalities and diverse emotion recognition objectives. Specifically, we propose a Brain anatomy-aware Inter-modal Hierarchical Graph Convolutional Network (BIH-GCN) with two key stages: (i) a local stage that aggregates EEG channels into anatomically defined cortical regions to capture region-specific dynamics, and (ii) a global stage that integrates these region-level representations with video and EEG embeddings through structured inter-modal message passing. In parallel, we introduce a correlation-based distribution alignment that normalizes covariance into correlations, reducing scale discrepancies between modalities while preserving their complementary variations. Taken jointly, these components move beyond naïve fusion by embedding neurophysiological priors and enabling structured, interpretable reasoning across EEG and video. As a result, our EVER not only improves robustness and generalization across heterogeneous modalities but also provides an interpretable framework that grounds EEG-video fusion in neurophysiological structure, paving the way for more reliable emotion recognition in various scenarios.

To systematically evaluate our framework, we establish an extensive benchmark on three public datasets including EAV (Lee et al., 2024), Emognition (Saganowski et al., 2022), and MD-MER (Yang et al., 2024). These datasets provide time-synchronized EEG–video recordings, essential for aligning observable video cues with neural dynamics in EEG. Moreover, we enforce subject-independent splits, ensuring that training and test sets do not share participants. This setup better reflects real-world deployment where unseen users are encountered and prevents subject-specific biases. To the best of our knowledge, no public benchmark has been explicitly designed for paired EEG–video emotion recognition. Further details of these datasets are provided in Sec. 4.1.

Our evaluation encompasses both unimodal and multi-modal settings and covers a total of 12 representative models. For the video modality, we benchmark 5 transformer-based video backbones (Arnab et al., 2021; Bertasius et al., 2021; Liu et al., 2021; Tong et al., 2022; Bandara et al., 2023). These models were not originally designed for emotion recognition but serve as strong baseline architectures for video classification tasks. For EEG, we evaluate 4 transformer-based architectures for sequence modeling (Zhang & Yan, 2023; Nie et al., 2023; Liu et al., 2024b; Wang et al., 2024) together with a Mamba-based state-space network (Erol et al., 2024). Finally, for the multi-modal setting, we include 2 existing audio–video emotion recognition networks (Tang et al., 2022; Sun et al., 2024), which provide a point of comparison against our proposed EEG–video fusion framework. This choice is motivated by the similarity between EEG and audio signals, as well as the use of strong transformer-based backbones. Extensive benchmarking demonstrates that our proposed framework consistently achieves state-of-the-art performance across three public datasets.

## 2 RELATED WORK

### 2.1 VIDEO AND EEG EMOTION RECOGNITION

Video- and EEG-based emotion recognition have been widely investigated as unimodal tasks. For video, several studies design task-specific networks with facial expression or affective behavior analysis (Zhang et al., 2019; 2021; Ben et al., 2021), but these approaches are not grounded in general-purpose backbones and often lack robustness in challenging conditions. Similarly, EEG-based methods typically introduce customized sequence or graph architectures (Gao et al., 2022; Du et al., 2022; Pan et al., 2023), yet they do not leverage strong backbone models, remaining sensitive to inter-subject variability and fixed electrode layouts that hinder adaptation across datasets.

This limitation motivates the adoption of general backbone architectures, which have transformed representation learning in vision and time-series domains but have been largely overlooked in the field of emotion recognition. To establish a fair benchmark, we adapt representative architectures to each modality, including ViViT (Arnab et al., 2021), Timeseriesformer (Bertasius et al., 2021), Swin Transformer (Liu et al., 2021), VideoMAE (Tong et al., 2022), and AdaMAE (Bandara et al., 2023) for video, as well as PatchTST (Nie et al., 2023), Crossformer (Zhang & Yan, 2023), iTransformer (Liu et al., 2024b), Medformer (Wang et al., 2024), and AudioMamba (Erol et al., 2024) for EEG. Although not originally designed for emotion recognition, these backbones provide a robust and unbiased basis for evaluating unimodal performance. In the multi-modal setting, we adopt TVLT (Tang et al., 2022) and HicMAE (Sun et al., 2024), developed for audio–video emotion recognition. Since audio signals share a similar data format with EEG, these models serve as meaningful

baselines in our benchmarking. However, their fusion strategies rely on naïve concatenation, which overlook modality-specific characteristics and fail to capture inter-modal dependencies.

To overcome this limitation, we introduce a framework that integrates a Brain anatomy-aware Inter-modal Hierarchical GCN (BIH-GCN) with a correlation-based distribution alignment. The BIH-GCN embeds anatomical priors and backbone embeddings into a unified graph to enable structured inter-modal reasoning, while the alignment module transforms covariance into correlations to reconcile feature distributions across two modalities. These components advance beyond naïve fusion and establish a principled foundation for generalizable EEG–video emotion recognition framework.

## 2.2 GRAPH CONVOLUTIONAL NETWORK

Graph Convolutional Network (GCN) (Kipf & Welling, 2017) is originally proposed to perform convolution-like operations on graph-structured data by aggregating neighbor node representations. In EEG-based emotion recognition, GCN have been employed to exploit spatial dependencies among electrodes by treating EEG channels as nodes and defining edges based on distances or anatomical priors (Gao et al., 2022; Du et al., 2022; Pan et al., 2023). Several studies have further introduced hierarchical GCNs that aggregate channel-level features into region representations. HD-GCN (Ye et al., 2022) proposed multi-level spatial dependencies via dual-branch modeling of global and local connectivity. PGCN (Jin et al., 2024) aggregated features at local, mesoscopic, and global scales informed by prior brain region definitions. MS-GCN (Du et al., 2022) introduced multi-scale relationships among channels and regions to enhance discriminative EEG representations.

However, existing GCN methods remain limited in two aspects. First, while these methods aggregate EEG signals from individual channels into coarse brain regions, this design remains confined to local grouping and fails to capture interactions between region-level representations and higher-level features learned by backbone networks. Second, these approaches rely on fixed brain-region definitions, constraining their applicability across datasets and ultimately limiting generalization. To address these limitations, we propose BIH-GCN, which enables inter-region information exchange through a two-stage GCN that first models intra-region dynamics and then captures inter-region interactions, while also integrating high-level global embeddings from EEG and video. For anatomical grounding, we adopt a standard five-region partition of the scalp (*i.e.*, Frontal, Temporal, Central, Parietal, and Occipital), following the international 10–20 EEG system in neuroscience (Herwig et al., 2003; Macorig et al., 2021). This ensures robustness across different EEG devices and electrode layouts. Our two-stage design enables structured inter-modal reasoning within a unified graph, moving beyond EEG-only region aggregation and ensuring robustness.

## 3 METHOD

### 3.1 OVERALL FRAMEWORK

Figure 1 illustrates our EEG-Video Emotion Recognition (EVER) framework. Our framework is designed to effectively unify EEG and video. In detail, it consists of the following three components.

**Video Network** We adopt AdaMAE (Bandara et al., 2023) to extract video representations from input frame sequences. Each video is uniformly sampled in time to obtain 32 frames, following prior studies (Arnab et al., 2021; Liu et al., 2021). The resulting video input is denoted as $V \in \mathbb{R}^{32 \times 3 \times H \times W}$, where $H$ and $W$ are the height and width of the video, respectively. We utilize the pretrained AdaMAE encoder trained on VoxCeleb (Nagrani et al., 2017) and discard the decoder, extracting the video embedding feature $\mathbf{z}_v$. Formally, the video embedding is obtained as follows:

$$\mathbf{z}_v = f_v(V) \in \mathbb{R}^d, \tag{1}$$

where $f_v$ denotes the AdaMAE encoder function and $d$ is the dimensionality of the embedding. This approach allows us to leverage rich spatiotemporal features learned from large-scale pretraining while providing compact and consistent representations suitable for fusion with EEG.

**EEG Network** To extract EEG representations corresponding to each video sample, we first transform raw EEG signals into a time–frequency representation using the Short-Time Fourier Transform (STFT), and then employ a state-space backbone for feature extraction. Let the input EEG for a sample be $\mathbf{E} \in \mathbb{R}^{C \times T_{\text{total}}}$, where $C$ is the number of EEG channels and $T_{\text{total}}$ denotes the total length of

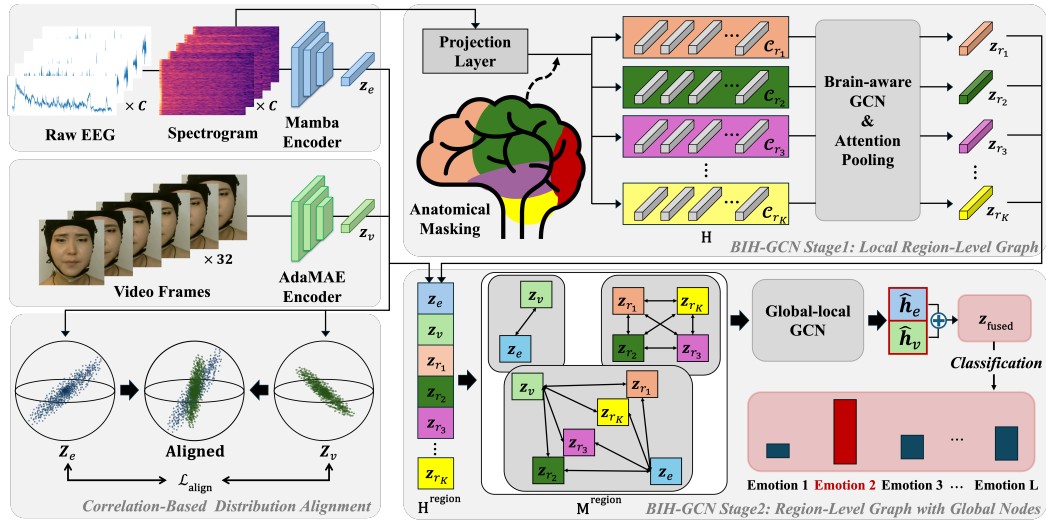

Figure 1: **Overall framework of EEG-Video Emotion Recognition (EVER).** Embeddings $\mathbf{z}_e$ and $\mathbf{z}_v$ are extracted from synchronized spectrogram and video, with $\mathcal{L}_{\text{align}}$ minimizing their discrepancy. In BIH-GCN stage 1, the spectrogram projection yields local nodes $\mathbf{H}$, which perform brain-aware GCN with anatomical masking within region $\mathcal{C}_{r_k}$ and attention pooling to produce region features $\mathbf{z}_{r_k}$. In stage 2, we form $\mathbf{H}^{\text{region}} = [\mathbf{z}_e, \mathbf{z}_v, \mathbf{z}_{r_1}, \dots, \mathbf{z}_{r_K}]^{\top}$ and apply global–local GCN with mask $\mathbf{M}^{\text{region}}$. Finally, emotion is predicted from the fused logit $\mathbf{z}_{\text{fused}}$ derived from $\hat{\mathbf{h}}_e$ and $\hat{\mathbf{h}}_v$.

the EEG signal. We compute the STFT for each channel with a window length of $n_{\text{fft}}$ and Hann window function (Zafar et al., 2022) as follows:

$$\mathbf{S}_c = \text{STFT}(\mathbf{e}_c, n_{\text{fft}}) \in \mathbb{C}^{F_{\text{raw}} \times L_{\text{raw}}}, \quad c \in [1, C], \tag{2}$$

where $\mathbf{e}_c \in \mathbb{R}^{T_{\text{total}}}$ is the raw signal of the $c$-th channel of $\mathbf{E}$, $F_{\text{raw}}$ is the number of frequency bins, and $L_{\text{raw}}$ is the number of frames determined by the window length. Next, we compute the magnitude and apply logarithmic scaling to compress the dynamic range as:

$$\mathbf{M}_c = \log\left(1 + |\mathbf{S}_c|\right) \in \mathbb{R}^{F_{\text{raw}} \times L_{\text{raw}}}, \quad c \in [1, C]. \tag{3}$$

Since the total sequence length differs between samples, each spectrogram $\mathbf{M}_c$ is interpolated to a fixed size of $F \times T$ along the frequency and time dimensions. By stacking the interpolated spectrograms across all channels, we obtain a unified EEG representation $\tilde{\mathbf{E}} \in \mathbb{R}^{C \times F \times T}$. Finally, $\tilde{\mathbf{E}}$ is passed through an EEG encoder $f_e$ to obtain EEG embedding $\mathbf{z}_e$:

$$\mathbf{z}_e = f_e(\tilde{\mathbf{E}}) \in \mathbb{R}^d. \tag{4}$$

In our framework, we instantiate $f_e$ with AudioMamba (Erol et al., 2024), as its selective state-space modeling aligns well with the non-stationary and long-range temporal dependencies of EEG signals. Moreover, the spectrogram of EEG exhibits structural similarity to audio spectrograms. This allows AudioMamba to transfer its design philosophy effectively to EEG, capturing local spectral variations as well as global temporal dynamics in a scalable manner.

**Fusion and Alignment Module** After obtaining the unimodal embeddings, we employ a BIH-GCN to model structured relationships among EEG channels and capture interactions between the two modalities. Prior studies (Maffei et al., 2023; Zhou et al., 2023) have shown that distinct brain regions respond differently to emotional stimuli, underscoring the importance of anatomical organization in EEG signals. Building on this insight, our fusion module leverages brain anatomy for representation learning and further extends it by hierarchically integrating inter-region interactions with high-level global embeddings from modality networks. In the first stage, channel-wise features are grouped according to predefined brain regions, and GCN is applied to capture intra-region relationships, producing region-level representations. In the second stage, these region-level features are integrated with $\mathbf{z}_v$ and $\mathbf{z}_e$, and a GCN models inter-modal interactions while directly producing classification logits. This hierarchical design respects the anatomical organization of the EEG

while enabling interpretable multi-modal fusion. To further reduce the modality gap, we introduce a correlation-based distribution alignment objective that encourages the similarity between the latent space distributions of the EEG and video embeddings. This alignment ensures that the fused features not only capture complementary information but are also represented in a shared latent space.

## 3.2 Brain Anatomy-aware Inter-modal Hierarchical GCN

Given the raw EEG spectrogram $\tilde{\mathbf{E}} \in \mathbb{R}^{C \times F \times T}$, we flatten each spectrogram channel and project it into a $d$-dimensional embedding space via flatten function $\text{flatten}(\cdot)$ and a linear layer $\text{proj}(\cdot)$ as:

$$\mathbf{H} = \text{proj}(\text{flatten}(\tilde{\mathbf{E}})) \in \mathbb{R}^{C \times d}, \tag{5}$$

where $\mathbf{H}$ is the channel-level embedding node matrix. We denote the $i$-th row of $\mathbf{H}$ as $\mathbf{h}_i \in \mathbb{R}^d$, corresponding to the embedding of the $i$-th channel.

**Stage 1: Local Region-level Graph** In the first stage, motivated by evidence that brain anatomy emphasizes region-specific interactions, we design a region-level graph that embeds anatomical priors and enables the network to capture intra-region relationships through GCN. Let $\mathcal{R} = \{r_k\}_{k=1}^K$ denote $K$ predefined brain regions. Each region $r_k$ is associated with a subset of EEG channels, represented by the index set $\mathcal{C}_{r_k} \subseteq \{1, \ldots, C\}$. Based on this anatomical prior, the local adjacency matrix $\tilde{\mathbf{A}}_{ij}$ is computed by cosine similarity with region-wise masking as follows:

$$\tilde{\mathbf{A}}_{ij} = \frac{\mathbf{h}_i^\top \mathbf{h}_j}{\|\mathbf{h}_i\| \, \|\mathbf{h}_j\|} \cdot \mathbf{M}_{ij}^{\text{local}}, \quad i, j \in [1, C]. \tag{6}$$

$$\mathbf{M}_{ij}^{\text{local}} = \begin{cases} 1, & \text{if } \exists k \text{ s.t. } i \in \mathcal{C}_{r_k} \text{ and } j \in \mathcal{C}_{r_k}, \\ 0, & \text{otherwise}, \end{cases} \tag{7}$$

where $\mathbf{M}^{\text{local}} \in \{0,1\}^{C \times C}$ encodes whether channels $i$ and $j$ belong to the same brain region $r_k$. After that, we apply symmetric normalization to balance node degrees as follows:

$$\hat{\mathbf{A}} = \mathbf{D}^{-1/2} \, \tilde{\mathbf{A}} \, \mathbf{D}^{-1/2}, \tag{8}$$

where $\mathbf{D}$ is the degree matrix of $\tilde{\mathbf{A}}$. A standard GCN layer (Kipf & Welling, 2017) is then applied over all EEG channels as follows:

$$\mathbf{H}^{\text{local}} = \text{GCN}(\mathbf{H}, \hat{\mathbf{A}}) \in \mathbb{R}^{C \times d}, \tag{9}$$

where $\mathbf{H}^{\text{local}}$ denotes the region-aware channel-level representations after graph propagation. To obtain compact region-level representations, we group the channel outputs according to the anatomical partition $\mathcal{R}$ and apply attention pooling within each brain region $r_k$. Specifically, attention pooling assigns a learnable weight $\alpha_i$ to each channel representation $\mathbf{h}_i^{\text{local}}$ within a region using a parameter vector $\mathbf{w}$, and aggregates them into a single region-level embedding $\mathbf{z}_{r_k}$ as follows:

$$\alpha_i = \frac{\exp(\mathbf{w}^\top \mathbf{h}_i^{\text{local}})}{\sum_{j \in \mathcal{C}_{r_k}} \exp(\mathbf{w}^\top \mathbf{h}_j^{\text{local}})}, \quad \mathbf{z}_{r_k} = \sum_{i \in \mathcal{C}_{r_k}} \alpha_i \, \mathbf{h}_i^{\text{local}} \in \mathbb{R}^d. \tag{10}$$

**Stage 2: Region-level Graph with Global Nodes** In the second stage, we regard the unimodal embeddings $\{\mathbf{z}_e, \mathbf{z}_v\}$ as global nodes and the region-level features $\{\mathbf{z}_{r_k}\}_{k=1}^K$ as local nodes. The overall node set is constructed as $\{\mathbf{z}_e, \mathbf{z}_v\} \cup \{\mathbf{z}_{r_k}\}_{k=1}^K$. After that, we stack these node embeddings to form the input node matrix for the region-level GCN as follows:

$$\mathbf{H}^{\text{region}} = [\mathbf{z}_e, \mathbf{z}_v, \mathbf{z}_{r_1}, \ldots, \mathbf{z}_{r_K}]^\top \in \mathbb{R}^{(2+K) \times d}. \tag{11}$$

Here, the adjacency matrix is explicitly defined by allowing only semantically valid connections. Instead of adopting a fully connected or purely similarity-driven graph, we incorporate domain knowledge through a structured binary mask: (i) the global video and EEG nodes are connected to model cross-modal interactions, (ii) each global node is linked to all region nodes to bridge global and local information, and (iii) inter-region connections are retained to reflect intra-brain dependencies. Formally, the binary mask $\mathbf{M}^{\text{region}}$ is constructed as follows:

$$\mathbf{M}_{ij}^{\text{region}} = \begin{cases} 1, & \text{if } (i,j) \in \{(\mathbf{z}_e, \mathbf{z}_v), (\mathbf{z}_e, \mathbf{z}_{r_k}), (\mathbf{z}_v, \mathbf{z}_{r_k}), (\mathbf{z}_{r_k}, \mathbf{z}_{r_{k'}})\}, \\ 0, & \text{otherwise}, \end{cases} \tag{12}$$

where $r_k, r_{k'} \in \mathcal{R}$ and $k \neq k'$. Following the same strategy as Eq. (6), the region-level adjacency matrix $\mathbf{A}^{\text{region}}$ is computed by cosine similarity, restricted to the valid entries of $\mathbf{M}^{\text{region}}$ for $i, j \in [1, 2+K]$ instead of using the local mask $\mathbf{M}^{\text{local}}$. Subsequently, symmetric normalization is applied in the same manner as Eq. (8) to obtain the normalized adjacency matrix $\hat{\mathbf{A}}^{\text{region}}$.

A second GCN layer is then applied to this region-level graph, enabling the model to integrate global video information, global EEG dynamics, and localized brain-region features into a unified representation. This layer directly projects each node into the label space as follows:

$$\hat{\mathbf{H}} = \text{GCN}\left(\mathbf{H}^{\text{region}}, \hat{\mathbf{A}}^{\text{region}}\right) \in \mathbb{R}^{(2+K) \times L}, \tag{13}$$

where $L$ denotes the number of target emotion classes. Among the updated node representations, the global nodes $\hat{\mathbf{h}}_e$ and $\hat{\mathbf{h}}_v$ correspond to the transformed embedding of $\mathbf{z}_e$ and $\mathbf{z}_v$, respectively. These nodes encode inter-modal correlations propagated through the global-level graph. The final fused feature is obtained by a weighted combination of the two global nodes as follows:

$$\mathbf{z}_{\text{fused}} = \omega \hat{\mathbf{h}}_e + (1 - \omega)\hat{\mathbf{h}}_v \in \mathbb{R}^L, \tag{14}$$

where $\omega$ denotes a learnable weight. The fused logit vector $\mathbf{z}_{\text{fused}}$ is subsequently used for classification without an additional classifier head. Note that we utilize the refined global embeddings to construct the final logit, balancing the visual and EEG modalities and further confining the role of region-level nodes to auxiliary and indirect support of global embedding enhancement.

Our two-stage hierarchical design respects the anatomical organization of the EEG by first modeling fine-grained channel interactions and then aggregating them into region-level features. It further enables inter-modal reasoning by integrating both unimodal embeddings and brain-region features.

## 3.3 Correlation-based Distribution Alignment

To effectively fuse the EEG and video modalities, it is crucial to align their feature distributions while preserving modality-specific variations. Therefore, we propose a correlation-based distribution alignment inspired by CORAL (Sun & Saenko, 2016), which enforces similarity between the second-order statistics of the unimodal embeddings. Let $\mathbf{Z}_e, \mathbf{Z}_v \in \mathbb{R}^{N \times d}$ denote the batch of unimodal embeddings, where $N$ is the batch size and $d$ is the feature dimension. After centering each batch by subtracting the mean, the covariance matrices are computed as follows:

$$\mathbf{\Sigma}_e = \frac{1}{N-1}\mathbf{Z}_e^\top \mathbf{Z}_e, \quad \mathbf{\Sigma}_v = \frac{1}{N-1}\mathbf{Z}_v^\top \mathbf{Z}_v, \quad \mathbf{\Sigma}_e, \mathbf{\Sigma}_v \in \mathbb{R}^{d \times d}. \tag{15}$$

Each entry captures the covariance between the corresponding feature dimensions. Next, the covariance matrices are converted into correlation matrices to quantify inter-feature dependencies as:

$$\mathbf{Corr}_{e,pq} = \frac{\mathbf{\Sigma}_{e,pq}}{\sqrt{\mathbf{\Sigma}_{e,pp}\mathbf{\Sigma}_{e,qq}}}, \quad \mathbf{Corr}_{v,pq} = \frac{\mathbf{\Sigma}_{v,pq}}{\sqrt{\mathbf{\Sigma}_{v,pp}\mathbf{\Sigma}_{v,qq}}}, \quad p, q \in [1, d], \tag{16}$$

where each entry $(p, q)$ captures the correlation between the $p$-th and $q$-th feature dimensions. Instead of directly aligning the covariances, we normalize them into correlation matrices to remove scale discrepancies and highlight their intrinsic dependency. To align the inter-feature dependencies between modalities, we minimize the squared differences of the off-diagonal elements as follows:

$$\mathcal{L}_{\text{align}} = \frac{1}{d(d-1)} \sum_{p=1}^{d} \sum_{q=1, q\neq p}^{d} \left(\mathbf{Corr}_{v,pq} - \mathbf{Corr}_{e,pq}\right)^2. \tag{17}$$

Minimizing $\mathcal{L}_{\text{align}}$ encourages unimodal embeddings to share similar correlation while preserving modality-specific characteristics, facilitating more coherent fusion in graph-based modeling.

## 3.4 Loss Functions

To ensure consistent comparison, we adopt the following unified loss for all benchmark models as:

$$\mathcal{L}_{\text{base}} = w_{\text{ce}}\mathcal{L}_{\text{ce}} + w_{\text{f}}\mathcal{L}_{\text{f}}, \tag{18}$$

where $\mathcal{L}_{ce}$ denotes the weighted cross-entropy loss (Cui et al., 2019) to mitigate class imbalance arising from participant's self-assessments, and $\mathcal{L}_f$ denotes the focal loss to further address hard-to-classify samples. Beyond this baseline formulation, our framework incorporates the proposed correlation-based distribution alignment. The final loss function is therefore defined as follows:

$$\mathcal{L}_{\text{total}} = \mathcal{L}_{\text{base}} + w_{\text{align}}\mathcal{L}_{\text{align}}. \tag{19}$$

## 4 EXPERIMENTAL RESULTS

### 4.1 IMPLEMENTATION DETAILS

**Datasets** We conduct experiments on three public datasets including MDMER (Yang et al., 2024), Emognition (Saganowski et al., 2022), and EAV (Lee et al., 2024), for which we collect paired EEG–video samples. **MDMER** dataset involves recordings from 73 participants across 32 emotion-eliciting video clips. From this dataset, we collect approximately 2.3K paired EEG–video samples, of which more than 1.8K (58 participants) are used for training and about 0.5K (15 participants) for validation. EEG was recorded with 18 channels, and videos were captured at 30 fps with $640 \times 480$ resolution. Ground-truth annotations were obtained from self-assessments of valence, arousal, and dominance on a 9-point scale. **Emognition** dataset contains recordings from 37 participants with affective multimedia stimuli. We construct 407 paired EEG–video samples, which are further split into 319 samples (29 participants) for training and 88 (8 participants) for validation. EEG was recorded with 4 channels, and videos were acquired at either 60 fps (297 samples) or 30 fps (110 samples) with $1080 \times 1920$ resolution. Self-assessments include valence, arousal, and motivation on a 9-point scale. **EAV** dataset consists of recordings from 37 participants with conversational interactions. For each subject, we collect 200 paired EEG–video samples, resulting in a total of 7.4K samples. We split the dataset into 5.8K samples (29 participants) for training and 1.6K (8 participants) for validation. Five discrete emotion categories are defined: Neutral, Anger, Happiness, Sadness, and Calmness, with EEG recorded using 30 channels and videos captured at 30 fps with $640 \times 480$ resolution. In our experiments, we evaluate emotion recognition along the valence and arousal dimensions for the MDMER and Emognition datasets. The original 9-point ratings are discretized into five categories by treating the midpoint as neutral: $\{1, 2\}, \{3, 4\}, \{5\}, \{6, 7\}, \{8, 9\}$. The EAV dataset is evaluated on five predefined emotion categories. Moreover, we construct brain regions $\mathcal{R}$ in Sec. 3.2 as five regions (*e.g.*, Frontal, Temporal, Central, Parietal, and Occipital) for MDMER and EAV datasets, and two regions (*e.g.*, Temporal and Frontal) for Emognition dataset, following the configuration of their EEG devices. As a preprocessing, facial regions from video are cropped using facial detection and resized to $224 \times 224$ across all datasets, following standard prior studies (Zafeiriou et al., 2017; Sun et al., 2024). Further dataset details are provided in A.3.

**Environments** All experiments are conducted under consistent settings to ensure fair comparison. For the benchmark evaluations, networks are trained for 100 epochs with a batch size of 16. To provide a more comprehensive evaluation, we report performance using not only accuracy but also unweighted average recall (UAR) (Sun et al., 2024) and weighted F1-score (W-F1) (Sharma, 2022). We set $w_{ce} = 0.9$ and $w_f = 0.1$ for all networks, and utilize $w_{align} = 0.5$ for our proposed $\mathcal{L}_{align}$. We employ the AdamW optimizer with $\beta = (0.9, 0.999)$, $\epsilon = 10^{-8}$. A cosine annealing warm restart schedule is used for learning rate adjustment. The implementation is based on PyTorch with CUDA 12.1 and executed on four NVIDIA RTX A6000 GPUs.

### 4.2 BENCHMARK RESULTS

Tables 1 and 2 present detailed benchmark results across the MDMER, Emognition, and EAV datasets, covering unimodal (*i.e.*, Video or EEG) and multi-modal (*i.e.*, Video+EEG) settings. All reported metrics are obtained under the same weight based on the best accuracy to ensure a fair comparison. On the MDMER dataset, AdaMAE achieves the second-best average accuracy of 39.2% with video, while iTransformer provides the highest UAR and second-best W-F1, and AudioMamba also attains an average accuracy of 39.2% with EEG. Since the MDMER clips average two minutes, the EEG modality is effective for capturing long-term emotional variations beyond the 32-frame limit of video. In the Emognition dataset, video models achieve average accuracy between 26.1% and 29.6%, whereas EEG models obtain higher accuracy ranging from 27.7% to 30.2%. However, TimeS achieves the second-highest UAR, and Swin records the second-best W-F1 with video. This indicates that while EEG networks are stronger in terms of average accuracy, video backbones yield superior W-F1 scores, reflecting their advantage under class imbalance. In the EAV dataset, TVLT achieves the second-best average accuracy of 42.0%. Overall, video backbones generally achieve stronger performance than EEG networks, particularly on class-balanced metrics. Specifically, video models yield W-F1 between 30.0%–45.5%, whereas EEG models obtain W-F1 of 9.1%–36.9%. In valence and arousal tasks, existing multi-modal networks (*i.e.*, HicMAE and TVLT) underperform strong unimodal baselines such as AdaMAE and AudioMamba, indicating that naïve fusion is less

Table 1: Performance comparison on the MDMER and Emognition datasets. We report Valence, Arousal, and their average score in terms of Accuracy (Acc), UAR, and W-F1.

| Modality | Model | Target | MDMER | | | Emognition | | |
|---|---|---|---|---|---|---|---|---|
| | | | Acc | UAR | W-F1 | Acc | UAR | W-F1 |
| Video | ViViT (Arnab et al., 2021) | Valence | 0.350 | 0.177 | 0.273 | 0.279 | 0.267 | 0.275 |
| | | Arousal | 0.354 | 0.253 | 0.264 | 0.244 | 0.211 | 0.213 |
| | | Average | 0.352 | 0.215 | 0.269 | 0.262 | 0.244 | 0.244 |
| | TimeS (Bertasius et al., 2021) | Valence | 0.419 | 0.289 | 0.342 | 0.284 | 0.291 | 0.379 |
| | | Arousal | 0.329 | 0.196 | 0.193 | 0.284 | 0.300 | 0.323 |
| | | Average | 0.374 | 0.243 | 0.268 | 0.284 | 0.296 | 0.351 |
| | Swin (Liu et al., 2021) | Valence | 0.398 | 0.201 | 0.309 | 0.273 | 0.308 | 0.434 |
| | | Arousal | 0.310 | 0.190 | 0.229 | 0.250 | 0.208 | 0.274 |
| | | Average | 0.354 | 0.196 | 0.269 | 0.262 | 0.258 | 0.352 |
| | VideoMAE (Tong et al., 2022) | Valence | 0.363 | 0.212 | 0.338 | 0.261 | 0.222 | 0.173 |
| | | Arousal | 0.310 | 0.184 | 0.200 | 0.261 | 0.225 | 0.285 |
| | | Average | 0.337 | 0.198 | 0.269 | 0.261 | 0.224 | 0.229 |
| | AdaMAE (Bandara et al., 2023) | Valence | 0.419 | 0.217 | 0.326 | 0.284 | 0.311 | 0.354 |
| | | Arousal | 0.365 | 0.227 | 0.229 | 0.307 | 0.200 | 0.117 |
| | | Average | 0.392 | 0.222 | 0.278 | 0.296 | 0.256 | 0.236 |
| EEG | PatchTST (Nie et al., 2023) | Valence | 0.417 | 0.201 | 0.249 | 0.247 | 0.205 | 0.198 |
| | | Arousal | 0.240 | 0.209 | 0.155 | 0.318 | 0.284 | 0.290 |
| | | Average | 0.329 | 0.205 | 0.202 | 0.283 | 0.245 | 0.244 |
| | iTransformer (Liu et al., 2024b) | Valence | 0.404 | 0.251 | 0.324 | 0.306 | 0.269 | 0.247 |
| | | Arousal | 0.365 | 0.267 | 0.296 | 0.247 | 0.219 | 0.225 |
| | | Average | 0.385 | **0.259** | 0.310 | 0.277 | 0.244 | 0.236 |
| | Crossformer (Zhang & Yan, 2023) | Valence | 0.423 | 0.218 | 0.266 | 0.318 | 0.262 | 0.272 |
| | | Arousal | 0.358 | 0.284 | 0.307 | 0.271 | 0.229 | 0.196 |
| | | Average | 0.391 | 0.251 | 0.287 | 0.294 | 0.246 | 0.234 |
| | Medformer (Wang et al., 2024) | Valence | 0.375 | 0.215 | 0.262 | 0.282 | 0.267 | 0.268 |
| | | Arousal | 0.317 | 0.223 | 0.209 | 0.282 | 0.292 | 0.289 |
| | | Average | 0.346 | 0.219 | 0.236 | 0.282 | 0.280 | 0.279 |
| | AudioMamba (Erol et al., 2024) | Valence | 0.425 | 0.263 | 0.292 | 0.273 | 0.200 | 0.084 |
| | | Arousal | 0.358 | 0.217 | 0.226 | 0.330 | 0.300 | 0.296 |
| | | Average | 0.392 | 0.240 | 0.259 | 0.302 | 0.250 | 0.190 |
| Video+EEG | TVLT (Tang et al., 2022) | Valence | 0.406 | 0.215 | 0.248 | 0.261 | 0.333 | 0.284 |
| | | Arousal | 0.350 | 0.192 | 0.227 | 0.239 | 0.217 | 0.229 |
| | | Average | 0.378 | 0.204 | 0.238 | 0.250 | 0.275 | 0.257 |
| | HicMAE (Sun et al., 2024) | Valence | 0.390 | 0.218 | 0.273 | 0.250 | 0.213 | 0.288 |
| | | Arousal | 0.354 | 0.200 | 0.174 | 0.307 | 0.200 | 0.117 |
| | | Average | 0.372 | 0.209 | 0.224 | 0.279 | 0.207 | 0.203 |
| | **EVER (Ours)** | Valence | 0.421 | 0.214 | 0.320 | 0.295 | 0.317 | 0.217 |
| | | Arousal | 0.406 | 0.264 | 0.301 | 0.375 | 0.400 | 0.492 |
| | | Average | **0.414** | 0.239 | **0.311** | **0.335** | **0.359** | **0.355** |

**Bold**: The best, Underline: The second-best

Table 2: Performance comparison on the EAV dataset. We report Accuracy (Acc) and W-F1 with the number of network parameters. Since each class is equally represented in EAV, UAR equals Acc.

| Modality | Model | Acc | W-F1 | Param (M) |
|---|---|---|---|---|
| Video | ViViT (Arnab et al., 2021) | 0.395 | 0.300 | 86.244 |
| | TimeS (Bertasius et al., 2021) | 0.304 | 0.311 | 121.108 |
| | Swin (Liu et al., 2021) | 0.369 | 0.383 | 58.730 |
| | VideoMAE (Tong et al., 2022) | 0.408 | 0.446 | 64.983 |
| | AdaMAE (Bandara et al., 2023) | 0.419 | 0.455 | 64.983 |
| EEG | PatchTST (Nie et al., 2023) | 0.209 | 0.137 | 5.775 |
| | iTransformer (Liu et al., 2024b) | 0.378 | 0.091 | 0.992 |
| | Crossformer (Zhang & Yan, 2023) | 0.358 | 0.345 | 12.760 |
| | Medformer (Wang et al., 2024) | 0.377 | 0.110 | 1.918 |
| | AudioMamba (Erol et al., 2024) | 0.418 | 0.369 | 0.008 |
| Video+EEG | TVLT (Tang et al., 2022) | 0.420 | 0.441 | 87.634 |
| | HicMAE (Sun et al., 2024) | 0.378 | 0.412 | 110.248 |
| | **EVER (Ours)** | **0.468** | **0.489** | 91.922 |

**Bold**: The best, Underline: The second-best

Table 3: Performance comparison of various combinations of our proposed methods on the MDMER and Emognition datasets. For each metric, we report the average score of valence and arousal.

| Method ($\mathcal{L}_{\text{base}}$) | $\mathcal{L}_{\text{align}}$ | BIH-GCN | MDMER | | | Emognition | | |
|---|---|---|---|---|---|---|---|---|
| | | | Acc | UAR | W-F1 | Acc | UAR | W-F1 |
| AdaMAE | | | 0.392 | 0.222 | 0.278 | 0.296 | 0.256 | 0.236 |
| AudioMamba | | | 0.392 | **0.240** | 0.259 | 0.302 | 0.250 | 0.190 |
| AdaMAE + AudioMamba | | | 0.373 | 0.207 | 0.250 | 0.261 | 0.258 | 0.199 |
| | ✓ | | 0.365 | 0.207 | 0.258 | 0.267 | 0.253 | 0.306 |
| | | ✓ | 0.405 | 0.215 | 0.274 | 0.324 | 0.317 | 0.296 |
| | ✓ | ✓ | **0.414** | **0.240** | **0.310** | **0.335** | **0.359** | **0.355** |

**Bold**: The best, Underline: The second-best

Table 4: Ablation study of the proposed BIH-GCN with respect to brain-region components on the MDMER and Emognition dataset.

| Method | w/o Stage1 $\mathbf{M}^{\text{local}}$ | w/o Stage2 $(\mathbf{z}_{r_k}, \mathbf{z}_{r_{k'}})$ | MDMER | | | Emognition | | |
|---|---|---|---|---|---|---|---|---|
| | | | Acc | UAR | W-F1 | Acc | UAR | W-F1 |
| Ours | | | **0.414** | **0.240** | **0.310** | **0.335** | **0.359** | **0.355** |
| | ✓ | | 0.402 | 0.231 | 0.303 | 0.313 | 0.350 | 0.354 |
| | | ✓ | 0.401 | 0.222 | 0.278 | 0.278 | 0.238 | 0.307 |
| | ✓ | ✓ | 0.383 | 0.205 | 0.242 | 0.278 | 0.231 | 0.287 |

w/o: without, **Bold**: The best, Underline: The second-best

Table 5: Ablation study of the feature alignment methods on the Emognition dataset.

| Method | Emognition | | |
|---|---|---|---|
| | Acc | UAR | W-F1 |
| MMD | 0.326 | 0.319 | 0.306 |
| CORAL | 0.318 | 0.321 | 0.308 |
| $\mathcal{L}_{\text{align}}$ | **0.335** | **0.359** | **0.355** |

**Bold**: Best, Underline: Second-best

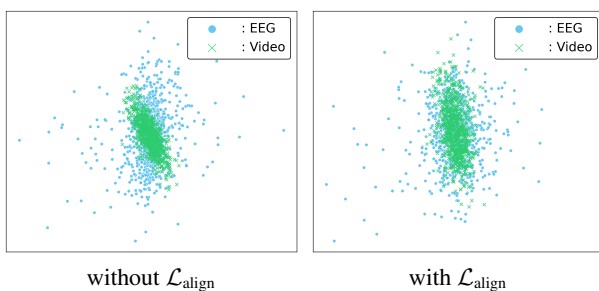

without $\mathcal{L}_{\text{align}}$     with $\mathcal{L}_{\text{align}}$

Figure 2: PCA visualization of latent space with and without $\mathcal{L}_{\text{align}}$ on the Emognition dataset.

effective than a single powerful backbone. To balance accuracy and efficiency, we adopt AdaMAE and AudioMamba as modality-specific backbones and utilize our proposed BIH-GCN. Our proposed framework achieves consistent state-of-the-art accuracy, outperforming the best among twelve baseline models by 2.2%p–4.8%p across all datasets. Moreover, our EVER attains the highest W-F1 and competitive UAR scores, while maintaining comparable parameters and achieving a balanced performance across all three metrics. Among multi-modal baselines, TVLT and HicMAE achieve Acc/W-F1 scores of 0.420/0.441 and 0.378/0.412 with 87.6M and 110.2M parameters, respectively, on the EAV dataset. In contrast, EVER attains markedly higher performance with 0.468 Acc and 0.489 W-F1 while utilizing only 91.9M parameters, demonstrating superior trade-offs between accuracy and efficiency. The fact that EVER consistently outperforms both heavy multi-modal and light single modal approaches indicates the clear benefit of jointly leveraging EEG and video information. These improvements stem from aggregating EEG channels into anatomically grounded region embeddings, applying structured graph reasoning where global nodes interact with local regions, and aligning EEG–video representations to better exploit complementary cues for robust emotion recognition. A detailed analysis of our proposed methods is provided in the following section.

## 4.3 EFFECTIVENESS OF THE PROPOSED FRAMEWORK

Table 3 compares different combinations of our proposed methods on the MDMER and Emognition datasets. While naïve fusion (*i.e.*, AdaMAE + AudioMamba) of EEG and video does not surpass unimodal networks, utilizing our BIH-GCN yields substantial improvements, with accuracy gains of 3.2%p on MDMER and 6.3%p on Emognition. Furthermore, correlation-based distribution alignment ($\mathcal{L}_{\text{align}}$) enhances graph learning by reducing discrepancies between heterogeneous

Table 6: Performance comparison of different combinations with other backbones.

| Method | MDMER | | | Emognition | | | EAV | |
|---|---|---|---|---|---|---|---|---|
| | Acc | UAR | W-F1 | Acc | UAR | W-F1 | Acc | W-F1 |
| ViViT | 0.352 | 0.215 | 0.269 | 0.262 | 0.244 | 0.244 | 0.395 | 0.300 |
| VideoMAE | 0.337 | 0.198 | 0.269 | 0.261 | 0.224 | 0.229 | 0.408 | 0.446 |
| Crossformer | 0.391 | 0.251 | 0.287 | 0.294 | 0.246 | 0.234 | 0.358 | 0.345 |
| ViViT + Crossformer + Ours | 0.401 | 0.211 | 0.259 | 0.318 | 0.263 | 0.344 | 0.455 | 0.416 |
| VideoMAE + Crossformer + Ours | 0.401 | 0.202 | 0.245 | 0.307 | 0.258 | 0.357 | 0.459 | 0.429 |

Table 7: Ablation study of the hyperparameter regarding loss objectives on the Emognition dataset.

| Method | $w_{ce}$ | $w_f$ | $w_{align}$ | Emognition | | |
|---|---|---|---|---|---|---|
| | | | | Acc | UAR | W-F1 |
| Ours | 0.9 | 0.1 | 0.5 | **0.335** | **0.359** | **0.355** |
| | 0.7 | 0.3 | 0.5 | 0.295 | 0.258 | 0.342 |
| | 0.5 | 0.5 | 0.5 | 0.296 | 0.209 | 0.258 |
| | 0.9 | 0.1 | 0.1 | 0.318 | 0.321 | 0.308 |
| | 0.9 | 0.1 | 0.3 | 0.324 | 0.317 | 0.296 |
| | 0.9 | 0.1 | 0.7 | 0.313 | 0.311 | 0.288 |

**Bold**: The best, Underline: The second-best

embeddings, resulting in the highest accuracy among all metrics across both datasets. Table 4 further validates the role of our BIH-GCN components. Removing either the region masking ($\mathbf{M}^{local}$) in Stage 1 or the global inter-region connections ($\mathbf{z}_{r_k}, \mathbf{z}_{r_{k'}}$) in Stage 2 causes clear performance drops. The full two-stage design achieves the best overall performance, confirming that both local anatomical grouping and global region-level graph are essential for robust fusion. For qualitative visualization of regional activations, please refer to A.1.

Moreover, we perform PCA visualization of EEG and video embeddings (*i.e.*, $\mathbf{z}_e$ and $\mathbf{z}_v$) to validate our correlation-based distribution alignment loss ($\mathcal{L}_{align}$), as shown in Fig. 2. $\mathcal{L}_{align}$ enforces a shared subspace across modalities, mitigating modality discrepancy and facilitating BIH-GCN performance as shown in Tab. 3. Table 5 further validate the effectiveness of our alignment loss. We replaced it with two alternatives including Maximum Mean Discrepancy (MMD) and CORAL on the Emognition dataset. Both substitutions lead to performance drops across all metrics, confirming that our correlation-based alignment is more effective in addressing the heterogeneity between the two modalities.

To further verify that EVER is inherently backbone-agnostic, we conducted experiments by replacing the AdaMAE and AudioMamba with alternative backbones. As shown in Tab. 6, proposed framework consistently yields substantial performance gains regardless of the backbone choice. Notably, combining different backbone pairs (*e.g.*, ViViT + Crossformer, VideoMAE + Crossformer) with Ours (BIH-GCN and correlation-based alignment loss) results in clear improvements across all evaluation metrics. These findings confirm that EVER does not rely on any specific architectural property of a particular backbone and is broadly applicable to a wide range of backbones.

To provide an empirical justification for our hyperparameter choices, we conducted hyperparameter searching on the Emognition dataset as shown in Tab. 7. The results indicate that deviations from this configuration generally lead to superior performance across all evaluation metrics.

## 5 CONCLUSION

In this paper, we propose the EVER framework, which integrates brain anatomy-aware hierarchical graph reasoning with correlation-based distribution alignment, consistently outperforming existing networks. Furthermore, we conducted an extensive benchmark for EEG–video paired emotion recognition, covering three public datasets and twelve representative models. Future research may focus on refining video cues via precise facial landmark and addressing temporal alignment challenges between modalities. We believe our framework, benchmark, and analyses provide a solid foundation for advancing research in interpretable EEG-video paired emotion recognition.

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

# A   APPENDIX

This appendix provides (i) additional visualizations of attention patterns across brain regions to further validate our proposed Brain anatomy-aware Inter-modal Hierarchical Graph Convolution Network (BIH-GCN), (ii) implementation details of the benchmark setup, and (iii) detailed descriptions of the datasets used in our experiments. Please refer to the following sections for details.

## A.1   BRAIN MAP VISUALIZATION

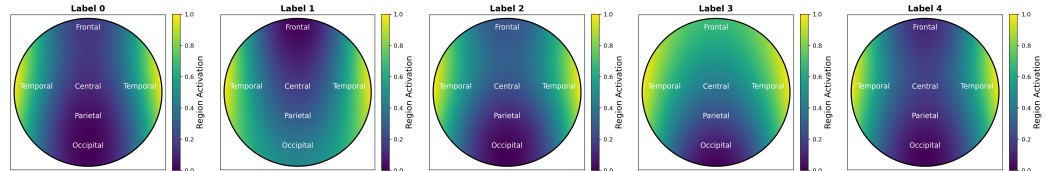

Figure A1: Average brain-region attention maps for five emotion categories: Neutral (label 0), Anger (label 1), Happiness (label 2), Sadness (label 3), and Calmness (label 4).

As shown in Fig. A1, we visualize the region-level brain activation maps for the five canonical areas (*i.e.*, Frontal, Temporal, Central, Parietal, and Occipital) on the EAV dataset (Lee et al., 2024). These maps are generated to illustrate the effectiveness of BIH-GCN in modeling regional interactions, with Stage 1 capturing intra-region dynamics and Stage 2 focusing on inter-region dependencies. By visualizing accumulated activation patterns across validation samples, we observe that different emotions elicit distinct regional activation distributions. This indicates that the two-stage design effectively leverages both local anatomical grouping and global region-level relationships. Specifically, modest emotions (*i.e.*, Neutral and Calmness) exhibit similar activations concentrated in the temporal region. In contrast, Anger induces elevated activation in the occipital region, consistent with increased posterior head tension commonly associated with heightened anger states. Moreover, Sadness shows stronger activations in the frontal and temporal regions, highlighting its link to affective regulation. This qualitative observation aligns with the quantitative results in Tab. 4, where removing either the region masking ($\mathbf{M}^{\text{local}}$) in Stage 1 or the global inter-region connections ($\mathbf{z}_{r_k}, \mathbf{z}_{r_{k'}}$) in Stage 2 leads to clear performance drops, and the full two-stage BIH-GCN achieves the best overall performance. Together, these results confirm that both intra-region and inter-region modeling are essential for robust brain-informed multi-modal fusion.

## A.2   IMPLEMENTATION DETAILS

Although we have already described the environment setup in Sec. 4.1 of the main manuscript, we provide additional details of the benchmark configuration. We employ the AdamW optimizer with $\beta = (0.9, 0.999)$, $\epsilon = 10^{-8}$. A cosine annealing warm restart schedule is used for learning rate adjustment. The initial learning rates are set to $2 \times 10^{-5}$ for transformer-based architectures (Arnab et al., 2021; Bertasius et al., 2021; Liu et al., 2021; Zhang & Yan, 2023; Nie et al., 2023; Liu et al., 2024b; Wang et al., 2024), $1 \times 10^{-4}$ for MAE-style foundation models (Tong et al., 2022; Bandara et al., 2023; Tang et al., 2022; Sun et al., 2024), and $2 \times 10^{-4}$ for Mamba model (Erol et al., 2024). The implementation is based on PyTorch with CUDA 12.1 and executed on four NVIDIA RTX A6000 GPUs.

## A.3   DETAILS OF DATASET

To further describe the details of the datasets, we visualize the 3D scatter plots of the self-assessment annotations for the Emognition (Saganowski et al., 2022) and MDMER (Yang et al., 2024) datasets as shown in Fig. A2. Each axis in the visualization corresponds to the respective annotation dimensions. The Emognition dataset is originally annotated along the dimensions of valence–arousal–motivation, while MDMER is based on valence–arousal–dominance. We observe that Emognition annotations are unevenly distributed across the space, whereas MDMER covers a

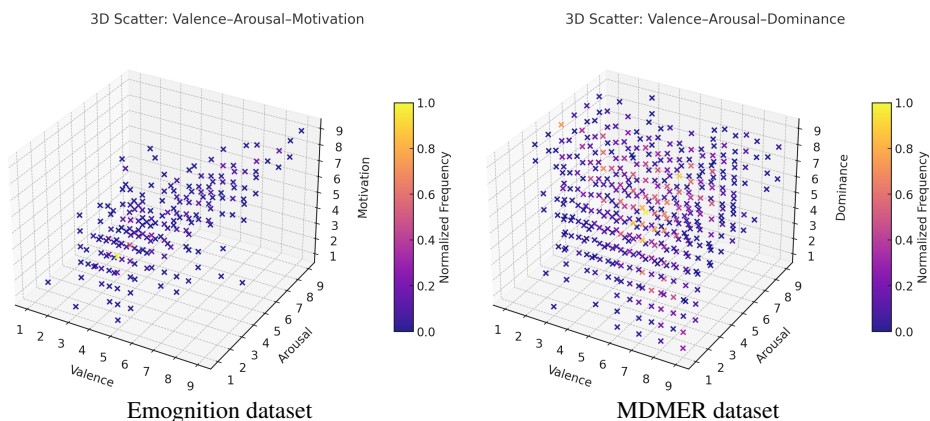

Figure A2: Distribution of self-assessment annotations (*i.e.*, ground-truth scores) on the Emognition and MDMER datasets.

wider range but remains concentrated near the middle scores. To address such imbalance and concentration issues during training, we adopt a combination of weighted cross-entropy loss and focal loss as described in Sec. 3.4. We focus our prediction on valence and arousal, which are the most widely adopted dimensions in emotion research (*e.g.*, the circumplex model of affect) and serve as a common, interpretable basis for evaluating emotional states. In the EAV dataset (Lee et al., 2024), each interaction lasted 20 seconds and was organized in paired Listen/Speak sessions. Since participants experienced 20 interactions for each of the five target emotions (*i.e.*, Neutral, Anger, Happiness, Sadness, and Calmness), the stimuli design ensures that all emotion categories are represented in equal proportions. Throughout our experiments, we excluded samples with missing EEG–video pairs.

Figures A3, A5, and A7 show representative video samples from the EAV, Emognition, and MD-MER datasets, while Figures A4, A6, and A8 present their corresponding EEG spectrograms across channels. Although these datasets contain salient affective stimuli (*e.g.*, sadness, disgust), visual cues alone often remain insufficient to accurately capture the underlying emotion. In contrast, EEG recordings directly reflect internal neural responses, offering complementary information about latent affective states. These observations underscore the necessity of combining EEG with video, thereby motivating our EEG-Video framework for emotion recognition. While the EEG signals vary across datasets due to differences in acquisition devices (*e.g.*, Muse 2), we apply a consistent Short-Time Fourier transform (STFT) to normalize the spectral representations, thereby enabling the network to learn in a consistent manner across heterogeneous recording setups. Please refer to our supplementary material for sample of videos.

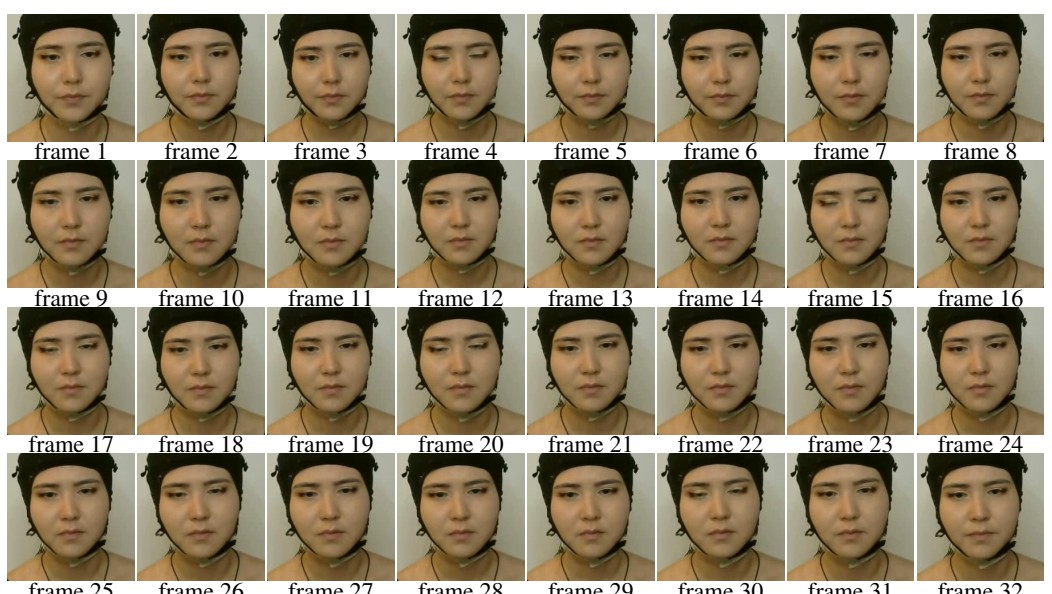

Figure A3: Sample of video frames on the EAV dataset with sad stimuli.

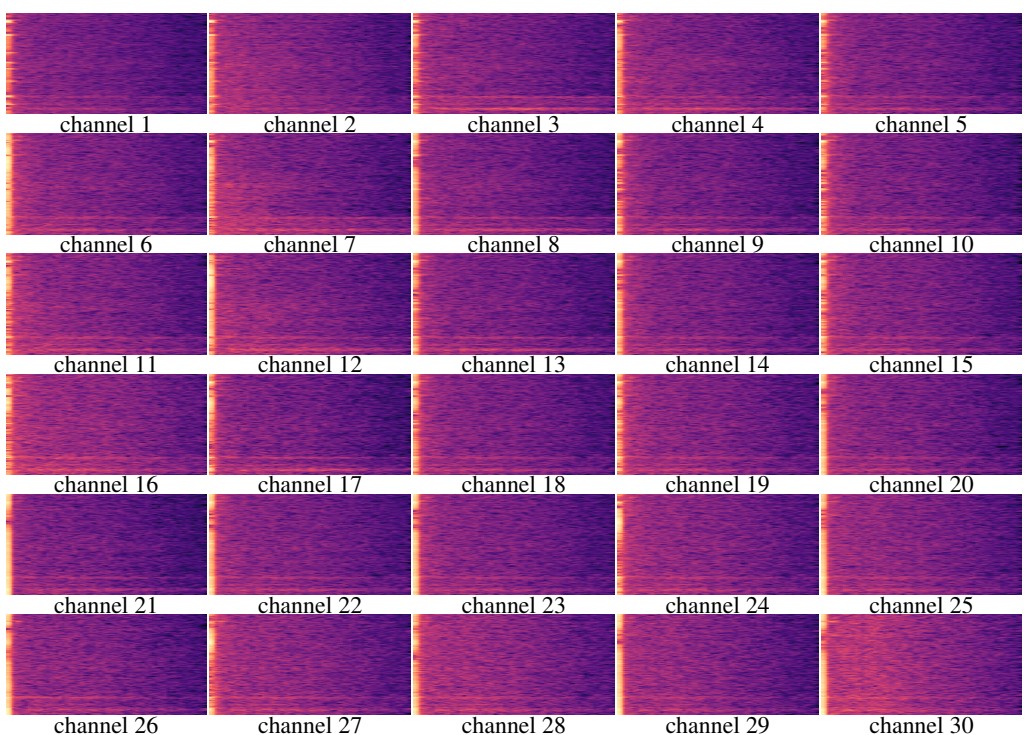

Figure A4: Sample of EEG spectrograms on the EAV dataset with sad stimuli.

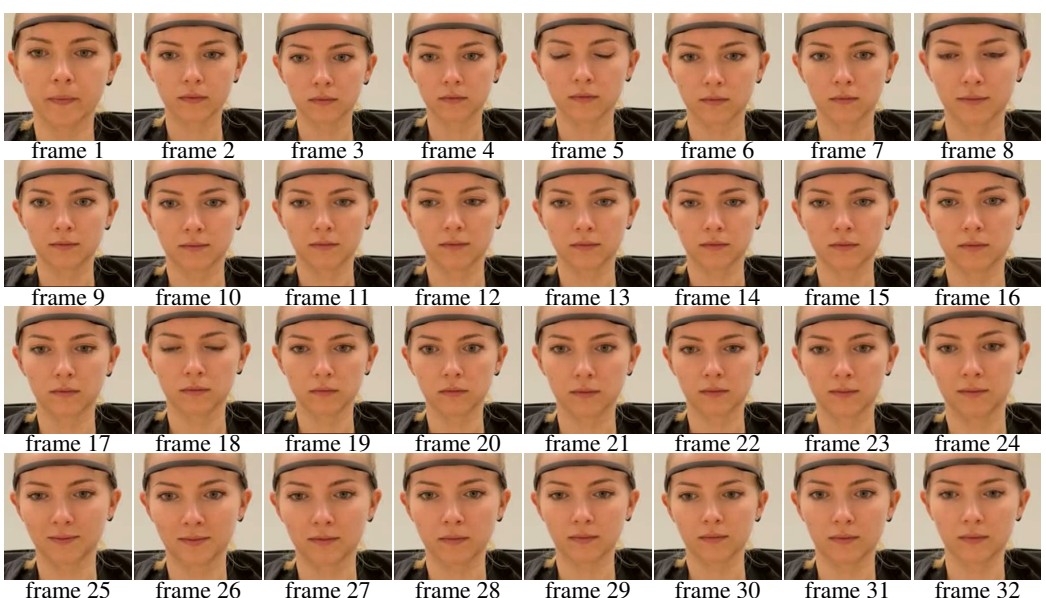

Figure A5: Sample of video frames on the Emognition dataset with disgust stimuli.

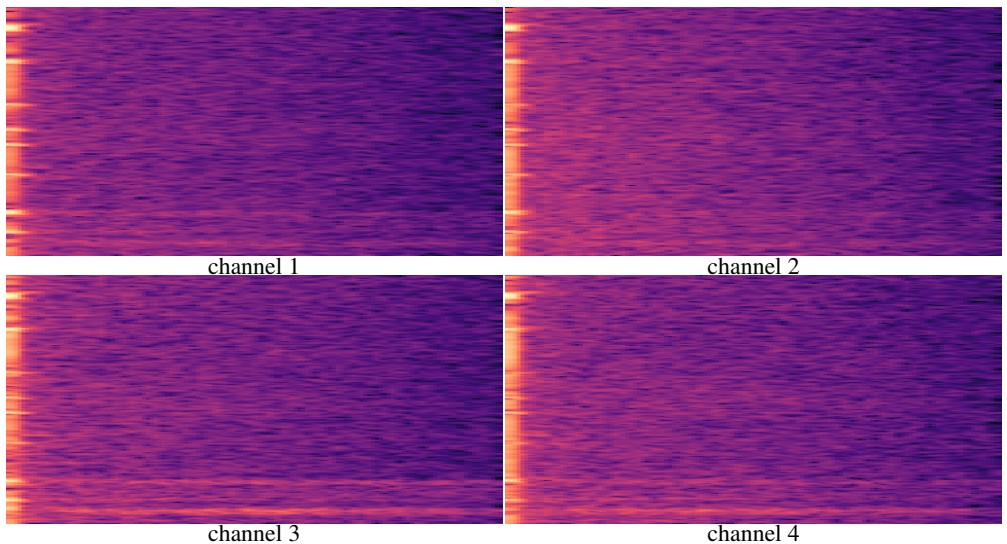

Figure A6: Sample of EEG spectrograms on the Emognition dataset with disgust stimuli.

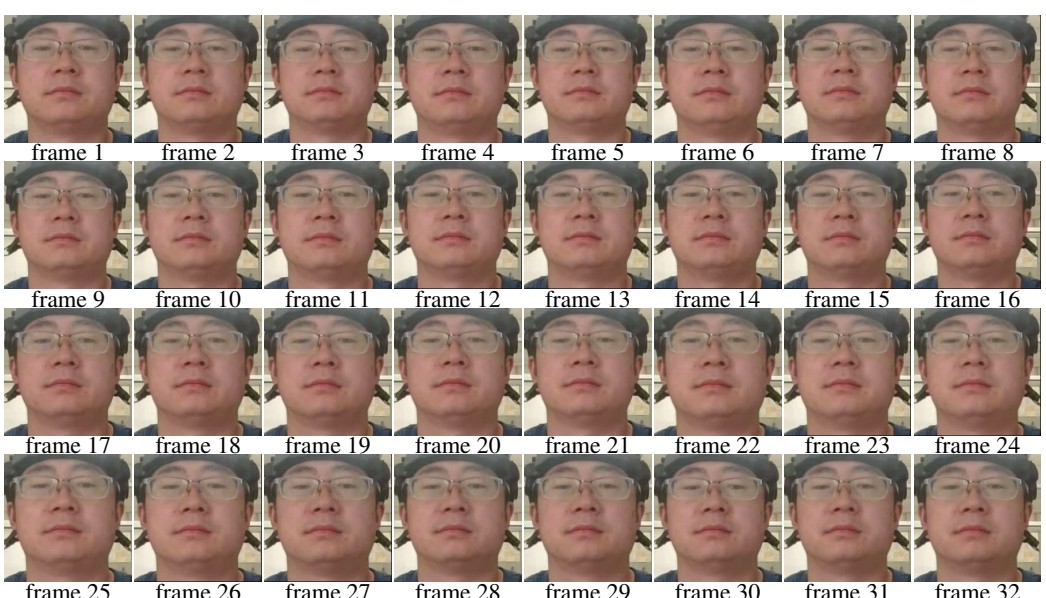

Figure A7: Sample of video frames on the MDMER dataset.

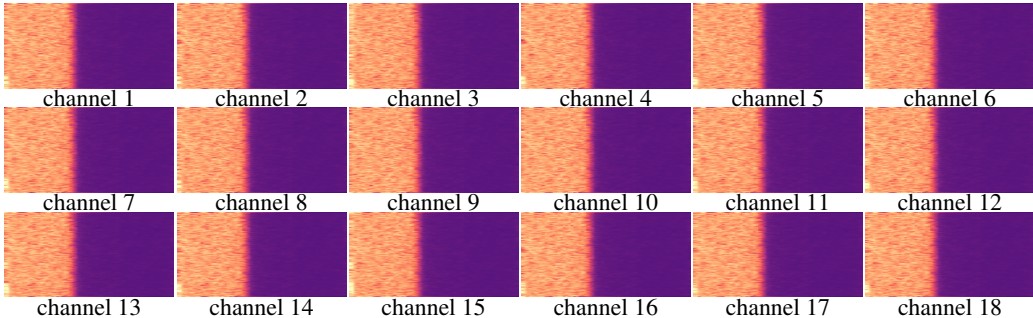

Figure A8: Sample of EEG spectrograms on the MDMER dataset.

