# OpenReview forum: "A Unified Framework for EEG–Video Emotion Recognition with Brain Anatomy Guidance"
_ICLR.cc/2026/Conference — Submitted to ICLR 2026_

### Official Review · Reviewer_wXbG · 2025-10-30

**Soundness:** 3
**Presentation:** 3
**Contribution:** 3
**Rating:** 6
**Confidence:** 4

**Summary:**

This paper propose a multi-modal emotion recognition framework EVER, which integrates EEG and video modalities to recognize emotional patterns. EVER employs a BIH-GCN module to combine region-level features with global embeddings for emotion classification. The framework is evaluated on three public EEG-video datasets, reporting improvements over previous works in emotion recognition task.

**Strengths:**

1. The authors introduce a multimodal framework for emotion classification, which, through extensive experiments, is shown to outperform existing unimodal approaches.
2. To mitigate the modality gap, this paper propose an alignment module which transforms covariance into correlations to reconcile feature distributions.
3. The paper presents its methodology with exceptional clarity. The hierarchical design of EVER is described in a structured, step-by-step manner.

**Weaknesses:**

1. The manuscript lacks sufficient detail and justification for certain hyperparameters. For example, the parameters $w_e,w_f, w_{align}$ are presented without any theoretical or empirical justification.
2. The brain region partitioning in BIH-GCN relies on a fixed anatomical mapping based on the standard 10–20 system, without accounting for individual anatomical variability or handling missing or sparse EEG channels. This rigid grouping may degrade performance in real-world scenarios where electrode placement varies across subjects or devices
3. The video branch employs AdaMAE pretrained on VoxCeleb—a dataset designed for speaker identification—which may bias the model toward identity-related facial attributes (e.g., hairstyle, skin tone) rather than affective cues such as subtle facial muscle movements, potentially limiting its sensitivity to genuine emotional expressions.

**Questions:**

1. In this paper the authors adpot AudioMamba, which is designed for audio, as the EEG encoder. Why not direct empoly an encoder specifically designed for EEG signal? How does the model perform when using raw EEG signals instead of the spectrogram employed in this paper?
2. Why are only the refined global embeddings used for final classification, while the region-level embeddings are excluded from the prediction head?

---

> ### Author Response · Authors · 2025-11-24
> **Response to Reviewer wXbG 's Weaknesses And Questions**
>
> We truly appreciate that the Reviewer-wXbG recognizes the novelty and practical significance of EVER, particularly highlighting BIH-GCN with its two-stage design. We also value your acknowledgment of the correlation-based distribution alignment, as well as your appreciation for the comprehensive experimental evaluation and the clarity and structured presentation of our methodology. Please kindly refer to the following responses addressing your concerns regarding the weaknesses and questions. Regarding the concern on AdaMAE pretraining, we will provide an additional comment and further clarification in a follow-up response as soon as possible.
>
> ---
>
> ### Weakness 1) Lack of theoretical justification for weights ($w_e, w_f, w_{align}$)
>
> We appreciate the reviewer’s comment regarding hyperparameter justification. As is common in multi-modal learning, hyperparameters related to loss terms do not require formal theoretical proof. Nevertheless, we recognize the importance of empirical validation. In the later response, we will provide results demonstrating the effect of varying each hyperparameter on performance, thereby offering a clear empirical justification for the chosen settings.
>
> ---
>
> ### Weakness 2) Brain-region grouping too rigid for real-world variability
>
> We clarify that under the standard 10–20 system, each EEG electrode has a fixed anatomical location, and the mapping from channels to regions is deterministic. The 10–20 system provides a standardized spatial partition covering the entire scalp, ensuring that all channels from different EEG devices can be consistently mapped and that missing or sparse of channels to regions does not occur.
>
> Moreover, the three datasets in our benchmark employ different device configurations, with 4, 18, and 30 channels respectively. EVER demonstrates strong and consistent performance across all of them, indicating that our design is robust to variations in channel density and electrode placement, even in low-density or wearable EEG scenarios.
>
> ---
>
> ### Question 1) Why AudioMamba for EEG? Why not use raw EEG?
>
> AudioMamba is suitable for our framework because EEG spectrograms share temporal–frequency structures with audio signals, allowing the model to effectively capture relevant patterns as mentioned in our main manuscript. Additionally, raw EEG signals vary in length across data samples due to inconsistent sampling rates and trial durations, which makes raw-signal processing challenging and potentially infeasible.
>
> Raw EEG signals would typically require extensive data augmentation and specialized architectures, such as MedFormer, which go beyond standard model training. Prior EEG studies also predominantly rely on spectrograms or other time–frequency transforms [1, 2, 3], so our approach aligns with established practices while providing a practical and efficient solution for multi-modal emotion recognition.
>
> ---
>
> ### Question 2) Why only use global embeddings for the final classifier?
>
> We thank the reviewer for this question. In our framework, the refined global embeddings are used for final classification because they already aggregate information from all region-level nodes through the hierarchical graph. Including region-level embeddings directly in the prediction head would introduce two issues. First, there is a modality imbalance: EEG has many more region nodes than video, which could bias the voting process. Second, there is a computational overhead, as including region-level features increases memory usage and inference cost. For these reasons, we rely on the global embeddings for the final prediction. In the revised version, we will explicitly clarify this design choice and discuss why only the global embeddings are used for classification.
>
> ---
>
> * [1] Wei-Long Zheng et al., Identifying stable patterns over time for emotion recognition from eeg. IEEE transactions on affective computing, 2017.
> * [2] Smith K Khare and Varun Bajaj. Time–frequency representation and convolutional neural network-based emotion recognition. IEEE transactions on neural networks and learning systems, 2020.
> * [3] Md Sakib Khan et al., Cnn-xgboost fusion-based affective state recognition using eeg spectrogram image analysis.Scientific Reports, 2022.

---

> ### Author Response · Authors · 2025-12-01
> **Ablation Study of Empirical Justification for Weights**
>
> To provide an empirical justification for our hyperparameter choices, we conducted additional experiments on the Emognition dataset by varying the weights $w_{ce}$, $w_{f}$, and $w_{align}$ in the EVER framework, beyond our default setting of $w_{ce} = 0.9$, $w_{f} = 0.1$, and $w_{align} = 0.5$.
>
> The results indicate that deviations from this configuration generally lead to superior performance across all evaluation metrics, confirming that our default weights represent an optimal balance for the loss terms in our method.
>
>
> | Method | $w_{ce}$ | $w_f$ | $w_{align}$ | Acc   | UAR   | W-F1  |
> |--------|------|-----|---------|-------|-------|-------|
> | Ours   | 0.9  | 0.1 | 0.5     | **0.335** | **0.359** | **0.355** |
> | Ours   | 0.7  | 0.3 | 0.5     | 0.295 | 0.258 | _0.342_ |
> | Ours   | 0.5  | 0.5 | 0.5     | 0.296 | 0.209 | 0.258 |
> | Ours   | 0.9  | 0.1 | 0.1     | 0.318 | _0.321_ | 0.308 |
> | Ours   | 0.9  | 0.1 | 0.3     | _0.324_ | 0.317 | 0.296 |
> | Ours   | 0.9  | 0.1 | 0.7     | 0.313 | 0.311 | 0.288 |
>
> **Bold = Best, Italic = Second-best**

---

### Official Review · Reviewer_95fJ · 2025-10-31

**Soundness:** 2
**Presentation:** 3
**Contribution:** 1
**Rating:** 2
**Confidence:** 4

**Summary:**

This paper proposes **EVER**, a unified EEG–video emotion recognition framework that integrates behavioral cues from video with neurophysiological information from EEG. The core contribution lies in a **brain anatomy-aware multimodal hierarchical graph convolutional network (BIH-GCN)**, which aggregates EEG channels into anatomically defined regions, combines them with global EEG and video embeddings, and performs structured cross-modal message passing. Furthermore, a **correlation-based distribution alignment loss** is introduced to reduce cross-modal discrepancies. Experimental results demonstrate that EVER achieves state-of-the-art performance.

**Strengths:**

The method is well-designed, integrating anatomical priors with multimodal fusion and alignment.
A comprehensive benchmark covering three datasets and twelve baselines is provided, with consistent evaluation metrics leading to the best results.

**Weaknesses:**

1. Novelty: The proposed method shows limited novelty. The BHI-GCN mentioned in the paper is a commonly used approach. Please clarify how Stage 1 of BIH-GCN differs from existing methods; and demonstrate the superiority of Stage 2 compared to fully connected or other GCNs that also consider both global and local connections. (e.g., [1], [2], [3])

2. Related Work:
   - The paper lacks references to existing EEG+video fusion methods, such as [4] and [5].
   - The GCN section contains some ambiguities: the proposed framework also relies on fixed brain region definitions and does not resolve cross-dataset applicability. For datasets with different EEG device layouts, the model still requires manual adjustment.

3. Experimental Section: The results only report mean values without standard deviations, making it difficult to assess the model’s stability under data distribution shifts.

4. The paper lacks comparison with dedicated EEG-based emotion recognition models.

5. It is recommended to include recent baselines from 2025 for fair comparison.

6. In the ablation study, it is suggested to add comparisons with different alignment methods.

[1] Wang J, Ning X, Xu W, et al. Multi-source Selective Graph Domain Adaptation Network for cross-subject EEG emotion recognition[J]. Neural Networks, 2024, 180: 106742.

[2] Jin M, Du C, He H, et al. PGCN: Pyramidal graph convolutional network for EEG emotion recognition[J]. IEEE Transactions on Multimedia, 2024, 26: 9070-9082.

[3] Ye M, Chen C L P, Zhang T. Hierarchical dynamic graph convolutional network with interpretability for EEG-based emotion recognition[J]. IEEE transactions on neural networks and learning systems, 2022.

[4] Jin X, Xiao J, Jin L, et al. Residual multimodal Transformer for expression‐EEG fusion continuous emotion recognition[J]. CAAI Transactions on Intelligence Technology, 2024, 9(5): 1290-1304.

[5]Lin N, Gao W, Li L, et al. vEpiNet: A multimodal interictal epileptiform discharge detection method based on video and electroencephalogram data[J]. Neural Networks, 2024, 175: 106319.

**Questions:**

1. Novelty: The proposed method shows limited novelty. The BHI-GCN mentioned in the paper is a commonly used approach. Please clarify how Stage 1 of BIH-GCN differs from existing methods ; and demonstrate the superiority of Stage 2 compared to fully connected or other GCNs that also consider both global and local connections.(e.g., [1], [2], [3])

2. Related Work:
   - The paper lacks references to existing EEG+video fusion methods, such as [4] and [5].
   - The GCN section contains some ambiguities: the proposed framework also relies on fixed brain region definitions and does not resolve cross-dataset applicability. For datasets with different EEG device layouts, the model still requires manual adjustment.

3. The experimental results present only mean values without standard deviations, making it impossible to assess the model’s stability under variations in data distribution.

4. Lack of appropriate baselines:
   - Add EEG+video fusion baselines for fair comparison.
   - Include dedicated EEG-based emotion recognition models as additional baselines.
5. In Table 2, EVER introduces a large increase in the number of parameters compared to the EEG-only models, yet the performance improvement is limited on some datasets. Please explain the rationale and significance behind this design choice.

6. Please explain why in Table 3 the combination of AdaMAE and AudioMamba results in lower performance, and why adding the alignment module yields a notable improvement only on Emognition (W-F1), while results on other metrics and datasets remain unchanged or even decline.

7. Ablation study:
   - Please include ablation results on the MDMER dataset in Table 4 to further verify the effectiveness of BIH-GCN.
   - To demonstrate the contribution of Stage 2 in BIH-GCN, it is recommended to add an additional ablation where the current three graphs are replaced by a single fully connected graph (treating the global EEG node, global video node, and all brain-region nodes as connected).

[1] Wang J, Ning X, Xu W, et al. Multi-source Selective Graph Domain Adaptation Network for cross-subject EEG emotion recognition[J]. Neural Networks, 2024, 180: 106742.

[2] Jin M, Du C, He H, et al. PGCN: Pyramidal graph convolutional network for EEG emotion recognition[J]. IEEE Transactions on Multimedia, 2024, 26: 9070-9082.

[3] Ye M, Chen C L P, Zhang T. Hierarchical dynamic graph convolutional network with interpretability for EEG-based emotion recognition[J]. IEEE transactions on neural networks and learning systems, 2022.

[4] Jin X, Xiao J, Jin L, et al. Residual multimodal Transformer for expression‐EEG fusion continuous emotion recognition[J]. CAAI Transactions on Intelligence Technology, 2024, 9(5): 1290-1304.

[5]Lin N, Gao W, Li L, et al. vEpiNet: A multimodal interictal epileptiform discharge detection method based on video and electroencephalogram data[J]. Neural Networks, 2024, 175: 106319.

---

> ### Author Response · Authors · 2025-11-24
> **Response to Reviewer 95fJ 's Weaknesses And Questions Part 1**
>
> We appreciate the Reviewer-95fJ’s effort in reading our submission. We regret if certain aspects of the EVER framework, particularly the design and role of BIH-GCN and the correlation-based alignment, were not fully clear. It seems that some misunderstandings may have led to concerns regarding novelty and design choices.
>
> Please kindly note that we have provided detailed responses to your questions and weaknesses below. We hope that reviewing these explanations will offer a more accurate understanding of our methodology and contributions.
>
> ---
>
> ### Weakness and Question 1) Novelty
>
> We would like to restate the essential clarification based on the distinction described in the main manuscript (lines 128–140).
>
> Crucially, the reviewer’s critique appears to conflate two different notions of “global” used in prior EEG GCN works versus our approach. Prior methods such as PGCN [1] and HD-GCN [2] (which we already cite and discuss) treat “global” as a pooled summary derived solely from channel-level EEG features. In contrast, in EVER we explicitly treat backbone-derived high-level embeddings (from both EEG backbones and video backbones) as global nodes that carry semantic, task-level information learned by large pretrained encoders.
>
> This is not a superficial change of terminology; it is an architectural and functional difference: Stage 1 of BIH-GCN models intra-region dynamics by aggregating channel-level signals into anatomically grounded region nodes, but we do so with design choices that preserve region-specific temporal patterns before higher-level fusion. Stage 2 does not simply pool region features; it performs structured inter-region message passing together with the backbone-derived global embeddings. This enables true inter-modal, hierarchical reasoning: region-region, region-global, and global-global exchanges that leverage the complementary semantics of deep backbone features and anatomically grounded EEG regions.
>
> By contrast, a fully-connected or naive GCN cannot exploit the semantic richness of pretrained backbones and is more prone to mixing low-level EEG noise with high-level visual semantics. Our two-stage design thus (i) preserves interpretable, anatomically motivated local representations, (ii) integrates robust, high-level modality semantics, and (iii) constrains cross-region interactions to meaningful pathways rather than unrestricted dense connectivity that can overfit or obscure modality-specific contributions.
>
> We have already cited and contrasted PGCN and HD-GCN in the manuscript and explained these differences. To remove any possible ambiguity, we will further clarify these distinct points in the revised manuscript.
>
> ---
>
> ### Weakness and Question 2) Related Work
>
> We appreciate the reviewer’s comments regarding the related work and the use of fixed brain-region definitions. While EEG-video fusion methods you mentioned were surveyed during the preparation of our manuscript, we did not include them as baselines because they employ non-standard video encoders (e.g., YOLO and ResNet), and both works are trained and evaluated on a single dataset without publicly available code. Consequently, these approaches cannot be reproduced or fairly compared within our benchmarking protocol. Nevertheless, we will cite them in the revised version for completeness.
>
> The comment on fixed brain-region definitions also reflects a misunderstanding. Under the standard 10–20 EEG system, each electrode has a fixed anatomical location, and the mapping from channels to cortical regions is deterministic. The 10–20 system provides a universal spatial partition covering the entire scalp, enabling consistent region assignments across devices with different channel densities. Moreover, our benchmark already includes datasets with markedly different device configurations (i.e., 4, 18, and 30 channels) and EVER shows strong and consistent performance across all of them. This demonstrates that our framework is robust to variations in channel density and electrode placement, achieving cross-dataset applicability.
>
> Finally, the reviewer suggests that our method requires manual adjustment across datasets, but this is not specific to our approach. In real-world scenarios, any EEG backbone or model must adapt to different numbers of channels when the device changes, because the input dimensionality itself varies. This is a fundamental property of EEG data rather than a limitation of our proposed framework. Our design, which maps channels to standardized brain regions defined by the 10–20 system, minimizes such adjustments and ensures maximum compatibility across heterogeneous EEG devices.
>
> ---
> * [1] Ming Jin et al., Pgcn: Pyramidal graph convolutional network for eeg emotion recognition. IEEE Transactions on Multimedia, 2024.
> * [2] Mengqing Ye et al., Hierarchical dynamic graph convolutional network with interpretability for eeg-based emotion recognition. IEEE Transactions on Neural Networks and Learning Systems, 2022.

---

> ### Author Response · Authors · 2025-11-24
> **Response to Reviewer 95fJ 's Weaknesses And Questions Part 2**
>
> ### Weaknesses 4, 5 and Question 4) Lack of Benchmark Comparisons
>
> We appreciate the reviewer’s suggestion regarding the inclusion of additional EEG-specific and EEG–video fusion baselines. However, we would like to clarify that our manuscript already establishes an extensive and highly competitive benchmark across 12 representative architectures evaluated on 3 EEG–video paired emotion recognition datasets.
>
> First, for the EEG modality, we benchmark transformer-based models that have been validated in recent top-tier literature, including PatchTST, Crossformer, iTransformer, MedFormer. In particular, MedFormer demonstrates strong performance with extensive EEG-specific benchmarks with these models for Alzheimer's disease, providing a rigorous and credible evaluation basis. Thus, our current setup already covers a broad spectrum of state-of-the-art EEG architectures that are widely adopted across sequential modeling domains. Nevertheless, we agree that including a few additional EEG-focused transformer-based models could further enrich the benchmark. We will therefore incorporate 2–3 more EEG-oriented transformer architectures to strengthen the unimodal comparisons in later response.
>
> Second, regarding multi-modal baselines, our manuscript deliberately adopts generalizable backbones and foundation models, rather than designing narrow EEG–video models. As emphasized in the main manuscript, this choice ensures fair, unbiased comparison and aligns with the recent trend in multi-modal foundation models. In line with the reviewer’s recommendation, we will additionally include recent 2025 audio–video foundation models.
>
> Please kindly note that, no prior work has conducted a benchmark of this scale, covering 12 networks across 3 datasets for EEG–video emotion recognition. Existing works typically evaluate only a small number of methods on a single dataset. Therefore, we respectfully note that our current benchmark already far exceeds the standard level of comparison in this research area, and we further enhance it with the additions mentioned above.
>
> ---
>
> ### Weakness 6 and Question 7) Ablation Studies
>
> We thank the reviewer for this helpful suggestion. We will include additional comparisons and clarifications in later response as follows: (1) We will extend the ablation study by including additional feature-alignment techniques such as CORAL. This result will be added as soon as possible. (2) We agree that including MDMER dataset results in Tab. 4 would further validate the effectiveness and robustness of BIH-GCN. We will conduct the ablation experiments on MDMER dataset and include the corresponding results.
>
> Moreover, the reviewer requests replacing our structured region–global graph with a single fully-connected graph that connects the global EEG node, global video node, and all K region nodes. Although our current formulation is densely connected, it intentionally incorporates a semantically constrained binary mask, preventing spurious edges and encoding domain knowledge (e.g., meaningful cross-modal links and inter-region brain dependencies). This mask is crucial for avoiding noisy similarity-driven edges.

---

> ### Author Response · Authors · 2025-11-24
> **Response to Reviewer 95fJ 's Weaknesses And Questions Part 3**
>
> ### Question 5) Number of Parameters
>
> We appreciate the reviewer’s observation. Below we clarify the rationale and significance behind the design choice. Although EEG-only models contain very few parameters, this is largely due to their domain-specific nature. EEG transformers are typically lightweight and can achieve strong performance with shallow architectures designed exclusively for low-dimensional. In contrast, the video backbones in our benchmark are selected for their generalization ability, robustness, and representational power across diverse real-world scenarios, which inevitably leads to a larger parameter budget.
>
> The key design rationale is that EVER is the unified framework that jointly models EEG and video signals for emotion recognition. Our goal is not to build the smallest possible framework, but to establish a general, extensible, and practically usable multi-modal foundation. In this context, integrating a strong video backbone is essential for leveraging complementary visual cues that EEG alone cannot capture.
>
> In terms of significance, the increase in parameters does not directly correspond to the complexity of our method. The proposed hierarchical reasoning module contributes only minimal overhead, and the overall parameter count remains competitive with existing multi-modal foundation models. Our proposed framework also achieves consistent state-of-the-art accuracy, outperforming the best among twelve baseline models by 2.2\%p–4.8\%p across all datasets. Moreover, our EVER attains the highest W-F1 and competitive UAR scores, while maintaining comparable parameters and achieving a balanced performance across all three datasets.
>
> ---
>
> ### Question 6) Regarding the Proposed Alignment
> We note that our proposed alignment method in EVER is not designed as a standalone improvement. Its effectiveness depends on the structured multi-level interactions learned by the BIH-GCN, which explicitly models relationships among global EEG, global video, and region-level EEG features.
>
> The contribution of the alignment module is intertwined with the global and regional feature reasoning. The gains emerge when the hierarchical structure is jointly trained, rather than from the alignment acting in isolation. This demonstrates that EVER’s performance improvements arise from the synergy between structured graph-based reasoning and correlation-based distribution alignment, rather than from alignment alone.

---

> ### Author Response · Authors · 2025-12-01
> **Further Benchmark Comparisons and Ablation Studies**
>
> We conducted additional benchmarks with EEG models and performed ablation studies about alternative feature alignment methods, as well as evaluated the effectiveness of our BIH-GCN on the MDMER dataset. Please refer to the details below.
>
> ---
>
> ### Weaknesses 4, 5 and Question 4) Further benchmark comparisons
>
> As discussed, our manuscript focuses on benchmarking general backbone architectures to provide a fair and broadly applicable evaluation. While we intended to include a recent audio–video foundation model, AVF-MAE [1],  the pretrained weights have not yet been publicly released, and thus we were unfortunately unable to evaluate it at this time.
>
> To further strengthen the EEG-specific networks, we incorporated two additional transformer-based models, Fedformer [2] and Informer [3], which have been validated in EEG-specific benchmarks [4]. With these additions, our benchmark now covers 14 representative architectures evaluated across 3 EEG–video paired emotion recognition datasets, providing an even more extensive and rigorous evaluation. This demonstrates the broad generality of our approach and ensures a comprehensive comparison across single and multi-modal architectures.
>
> | Method     | Acc \ (MDMER) | UAR \ (MDMER) | W-F1 \ (MDMER) | Acc \ (Emognition) | UAR \ (Emognition) | W-F1 \ (Emognition) | Acc \ (EAV) | W-F1 \ (EAV) |
> |------------|-----------|-----------|------------|----------------|----------------|----------------|---------|-----------|
> | Fedformer  | 0.292     | 0.206     | 0.195      | 0.294          | 0.282          | 0.292          | 0.256       | 0.251         |
> | Informer   | 0.392     | 0.252     | 0.282      | 0.300          | 0.272          | 0.247          | 0.226       | 0.221         |
>
> ---
>
> ### Weakness 6) Ablation study with other feature alignment methods
>
> To assess the effectiveness of our proposed feature alignment loss, we conducted experiments on the Emognition dataset by replacing our correlation-based alignment method ($L_{align}$) with two alternatives: CORAL, which inspired the design of our method, and MMD (Maximum Mean Discrepancy), a statistical measure used to align the distributions of two domains. The results show that both alternatives lead to noticeable performance drops across all metrics. These findings indicate that our tailored alignment loss is better suited for capturing the complex relationships between the two heterogeneous modalities.
>
> | Method                 | Acc            | UAR            | W-F1           |
> |------------------------|----------------|----------------|----------------|
> | MMD                    | 0.326   | 0.319          | 0.306          |
> | CORAL                  | 0.318          | 0.321    | 0.308    |
> | Ours ($L_{align}$)     | **0.335**      | **0.359**      | **0.355**      |
>
> **Bold = Best**
>
>
> ---
>
> ### Question 7) Ablation study on the MDMER dataset in Table 4
>
> To address the reviewer’s comment and provide further analysis of the stage-wise components of BIH-GCN, we conducted ablation experiments on the MDMER dataset. The results are consistent with those observed on the Emognition dataset: removing either stage 1 or stage 2 leads to a clear performance drop across all metrics. This consistency indicates that each stage contributes meaningfully to the overall model performance and further demonstrates that the proposed BIH-GCN framework is generally effective across different EEG device settings.
>
>
> | Method | w/o Stage1 | w/o Stage2 | Acc \ (MDMER) | UAR \ (MDMER) | W-F1 \ (MDMER) | Acc \ (Emognition) |  UAR \ (Emognition) | W-F1 \ (Emognition) |
> |--------|---------------------|---------------------------------|-----------|-----------|------------|----------------|----------------|----------------|
> | Ours   |                     |                                 | **0.414** | **0.240** | **0.310** | **0.335**      | **0.359**      | **0.355**      |
> | Ours   | ✓                   |                                 | _0.402_   | _0.231_   | _0.303_   | _0.313_        | _0.350_        | _0.354_        |
> | Ours   |                     | ✓                               | 0.401     | 0.222     | 0.278     | 0.278          | 0.238          | 0.307          |
> | Ours   | ✓                   | ✓                               | 0.383     | 0.205     | 0.242     | 0.278          | 0.231          | 0.287          |
>
> **w/o: without, Bold = Best, Italic = Second-best**
>
> ---
> [1] Wu et al., AVF-MAE++: Scaling Affective Video Facial Masked Autoencoders via Efficient Audio-Visual Self-Supervised Learning, CVPR, 2025
>
> [2] Zhou et al., Fedformer:Frequency enhanced decomposed transformer for long-term series forecasting, PMLR, 2022.
>
> [3] Zhou et al., Informer: Beyond efficient transformer for long sequence time-series forecasting, AAAI,
> 2021.
>
> [4] Wang et al. Medformer: A multi-granularity patching transformer for medical time-series classification, NeurIPS, 2024.

---

### Official Review · Reviewer_8y7d · 2025-11-01

**Soundness:** 3
**Presentation:** 3
**Contribution:** 3
**Rating:** 4
**Confidence:** 4

**Summary:**

The paper proposes EVER, a novel EEG–Video Emotion
Recognition framework that effectively integrates complementary information
from both modalities. Specifically, EVER employs a Brain anatomy-aware
Inter-modal Hierarchical Graph Convolution Network (BIH-GCN), which aggregates
EEG channel features into region-level representations guided by anatomical
priors. These region-level features are combined with global high-level EEG and
video backbone embeddings to form a unified representation for emotion classification.
Furthermore, the paper introduces a correlation-based distribution alignment loss
to reconcile modality-specific embeddings and reduce cross-modal discrepancies.
Extensive
experiments demonstrate that the proposed EVER achieves state-of-theart
performance by jointly modeling behavioral cues from video and physiological
responses from EEG, thereby enabling the recognition of emotional patterns
unattainable by either modality alone.

**Strengths:**

The paper introduces a novel EEG–Video Emotion Recognition (EVER) framework that explicitly
unifies video and EEG representations to overcome the limitations of existing multi-modal
architectures across heterogeneous modalities and diverse emotion recognition objectives. Specifically,
The paper proposes a Brain anatomy-aware Inter-modal Hierarchical Graph Convolutional Network
(BIH-GCN) with two key stages: (i) a local stage that aggregates EEG channels into anatomically
defined cortical regions to capture region-specific dynamics, and (ii) a global stage that integrates
these region-level representations with video and EEG embeddings through structured inter-modal
message passing. In parallel, the paper introduces a correlation-based distribution alignment that normalizes
covariance into correlations, reducing scale discrepancies between modalities while preserving
their complementary variations.

This paper has well realized multimodal sentiment recognition and demonstrates considerable innovation. Meanwhile, its experimental design is relatively comprehensive.

**Weaknesses:**

While the experiments in this paper are relatively comprehensive, there is a lack of discussion on certain comparative aspects. For instance, the AudioMamba algorithm, whose parameter count is less than one-thousandth of that of the algorithm proposed in this paper, still achieves performance superior to most other algorithms. The paper fails to provide an effective discussion on the balance between performance and computational complexity. It seems unreasonable to improve performance to a certain extent at the cost of computational efficiency.

Meanwhile, this paper integrates two existing methods to construct the baseline model, which leads to a decline in performance and makes it difficult to effectively demonstrate the advantages of the algorithm design.

**Questions:**

Please refer to the issues mentioned in the weaknesses section.

---

> ### Author Response · Authors · 2025-11-24
> **Response to Reviewer 8y7d 's Weaknesses**
>
> We truly appreciate that the Reviewer-8y7d recognizes the novelty and careful design of EVER, particularly highlighting our BIH-GCN’s hierarchical structure, the integration of EEG and video modalities, and the thoughtful use of correlation-based alignment. We also value your acknowledgment of the comprehensive experimental evaluation and the demonstrated effectiveness in capturing complementary emotional cues across modalities. Please kindly refer to the following responses addressing your concerns about computational complexity and performance decline of integrating two existing backbones.
>
> ---
>
> ### 1) No discussion of performance vs. computational complexity
>
> We appreciate the reviewer’s comment regarding the trade-off between performance and computational complexity. We kindly note that EEG-based backbones are inherently lightweight because they are designed for specific domains and can achieve strong performance with shallow architectures. In contrast, the video backbones used in our benchmark are selected for their generalization ability across diverse scenarios, which necessarily results in higher computational cost.
>
> As summarized in Table 2, multi-modal models such as TVLT and HicMAE achieve 0.420/0.441 and 0.378/0.412 on Acc/W-F1 with 87.6M and 110.2M parameters, respectively. EVER attains substantially higher performance (0.468 Acc and 0.489 W-F1) while requiring only 91.9M parameters. This demonstrates that our hierarchical GCN design introduces minimal computational overhead relative to existing multi-modal architectures.
>
> In addition, although AudioMamba is extremely lightweight at just 0.008M parameters, its single modal audio representation results in much lower accuracy (0.418 Acc and 0.369 W-F1). The fact that EVER significantly outperforms both heavy multi-modal baselines and light single modal approaches indicates the clear benefit of jointly leveraging EEG and video information. Importantly, EVER represents the first unified framework to model EEG and video together for emotion recognition, establishing a solid foundation for future multi-modal research.
>
> To further illustrate the necessity of combining EEG and video, in our later response we will include failure-cases: 1) video modality predicts correctly while the EEG modality does not, 2) the EEG modality predicts correctly while the video modality does not, and 3) the EEG+video correctly predicts the label while neither modality alone succeeds. We believe that these examples will provide clear evidence that integrating both modalities yields tangible improvements beyond what is possible with either modality in isolation.
>
> ---
>
> ### 2) Integrating two existing backbones induces performance decline
>
> Naively combining backbones from different modalities often leads to degradation performance in other multi-modal tasks [1, 2, 3] as well, due to inherent differences in task characteristics and modality-specific feature distributions. Table 3 in our main manuscript illustrates this phenomenon: simply fusion the EEG and video backbone outputs does not fully exploit complementary information between modalities.
>
> In contrast, EVER integrates these backbones through our proposed components including BIH-GCN and correlation-based distribution alignment that leverages both EEG and video signals effectively. As demonstrated in Tables 1 and 2, this integrated design consistently achieves state-of-the-art performance across all three benchmark datasets, highlighting the benefits of the unified architecture over a naive combination of separate backbones.
>
> ---
>
> * [1] Kailai Zhou et al., Improving multispectral pedestrian detection by addressing modality imbalance problems. In Proc. of European Conf. on Computer Vision (ECCV), 2020.
> * [2] Yunfeng Fan et al., Pmr: Prototypical modal rebalance for multimodal learning. In Proc. of Computer Vision and Pattern Recognition (CVPR), 2023.
> * [3] Zirun Guo et al., Classifier-guided gradient modulation for enhanced multimodal learning. Proc. of Advances in Neural Information Processing Systems (NeurIPS), 2024.

---

> ### Author Response · Authors · 2025-12-01
> **Failure Case Analyses and the Necessity of Multi-modal Integration**
>
> We observed situations where the video modality makes correct predictions while the EEG modality fails, and vice versa.
>
> EEG succeeds in scenarios involving subtle, transient emotional responses, such as brief internalized reactions or micro-level arousal changes, which are not clearly reflected in facial expressions or occur between video frames. In these cases, the video modality tends to misclassify emotions, whereas EEG can capture the underlying cognitive–affective dynamics. Conversely, the video modality succeeds when participants exhibit overt, behaviorally clear emotional expressions, but EEG signals are unreliable due to noise, motion artifacts, or poor electrode contact. Here, facial expressions provide stable cues, while EEG becomes ambiguous or inconsistent.
>
> These contrasting patterns underscore that neither modality alone is sufficient for robust emotion recognition. A structured fusion approach that effectively integrates EEG and video is therefore essential. By combining the complementary strengths of both modalities, our model is able to correctly predict labels in cases where neither EEG nor video alone would succeed. This demonstrates that the integration goes beyond merely increasing computational cost; it is a necessary step to capture the full spectrum of emotional information, highlighting the practical significance of EEG+video fusion.
>
> Importantly, EVER represents the first unified framework for EEG–video emotion recognition, establishing a foundation for future multi-modal research. We believe that our work provides a roadmap for the development of more robust, generalizable emotion recognition systems in future studies.

---

### Official Review · Reviewer_yFBv · 2025-11-02

**Soundness:** 3
**Presentation:** 2
**Contribution:** 2
**Rating:** 4
**Confidence:** 4

**Summary:**

The paper introduces EVER, an EEG and video emotion recognition framework that aims to make physiological signals and visual cues work in a single structured model. The authors argue that emotion recognition with only video or only EEG gives a partial view and is brittle across subjects and recording conditions, so they build a multimodal pipeline that extracts high level features from video with AdaMAE and from EEG spectrograms with AudioMamba, then fuses them through a two stage Brain anatomy aware Inter modal Hierarchical GCN. The first stage aggregates channel features into anatomically grounded regions through masked graph propagation and attention pooling. The second stage places both global modality embeddings and the region nodes in one graph and performs structured message passing so that video and EEG can exchange information while remaining connected to brain regions. A correlation based distribution alignment loss is added to reduce statistical mismatch between the two modalities. The authors further provide what is essentially a benchmark for EEG and video pair emotion recognition on three public datasets, namely MDMER, Emognition and EAV, and compare twelve representative unimodal and multimodal baselines. EVER gives the best overall numbers on the three datasets and the ablations show that both the hierarchical graph and the correlation loss are necessary.

**Strengths:**

The work targets a setting that is clearly under serviced. Existing affective computing studies often focus on video alone or on EEG alone, while audio and video fusion is better studied. Positioning the model around paired EEG and video recordings is therefore meaningful for the ICLR audience that is interested in multimodal learning and in models that can work with physiological signals. The model design is careful. The first graph stage does not simply connect electrodes according to similarity. It enforces a mask derived from standard scalp regions and then performs attention pooling inside each region, which makes the subsequent fusion less sensitive to the specific channel layout of a dataset. This is a sound way to bring domain priors into a modern backbone. The second graph stage is the part that actually unifies the modalities. By treating the video embedding, the EEG embedding and the region representations as nodes in the same graph, the model creates an explicit route for inter modality reasoning instead of the usual concatenation that appears in audio and video models. This is consistent with the motivation stated in the introduction. The distribution alignment objective is a small but useful addition. It operates on correlations rather than raw covariance and therefore avoids scale problems. The ablations confirm that alignment alone does not solve the task but in combination with the anatomy aware graph it brings the model over both unimodal baselines and over naive fusion. The experimental section is richer than what is typical for a first paper on a new multimodal pairing. The authors collect three public datasets that have time aligned EEG and video, they impose subject independent splits, and they report accuracy, unweighted average recall and weighted F1, so the reader can interpret results under both class balanced and class imbalanced regimes. On all three datasets EVER improves over the strongest single modality model, which is the right standard to use in multimodal work. The paper is technically self contained. The construction of the two adjacency masks, the attention pooling, the fusion of the two global nodes into logits and the formulation of the alignment loss are all given in sufficient detail so that an experienced reader can implement the model. The writing is clear and connects the graph construction to actual neuroscience practice on five scalp regions.

**Weaknesses:**

Although the paper claims a unified framework, in practice the current system is anchored to a very specific choice of backbones. Video features come from AdaMAE and EEG features come from a spectrogram that is processed by AudioMamba. The paper does not test the framework with lighter or more conventional EEG encoders such as shallow CNNs or temporal transformers trained from scratch, nor does it show whether the hierarchical graph can compensate when one modality is considerably weaker. This limits the generality of the claim. The benchmark is valuable but the size of the paired data is still modest. Emognition has slightly more than four hundred pairs. MDMER has about two point three thousand pairs. EAV is larger but its labels are discrete and balanced. Generalization to truly large scale recordings, to unlabeled synchronized streams, or to field recordings with missing frames is not studied. The model relies on discrete brain regions derived from the 10 20 system and maps channels to these regions through a fixed mask. This is reasonable for the three datasets at hand, yet cross dataset variation in channel count is already visible. The paper partially alleviates this through attention pooling, yet it does not quantify how sensitive the model is to errors in the region assignment or to setups with very few channels such as six or fewer electrodes which are common in wearable EEG devices. The correlation based alignment is motivated as a way to reduce modality discrepancy, yet there is no comparison with other simple alignment objectives such as maximum mean discrepancy or contrastive matching of the global embeddings. The current ablation only shows with and without. A stronger analysis would include parameter free alternatives to confirm that choosing correlations is important. The interpretation part could go further. Figure A1 shows that different emotions activate different regions, which is interesting, but the main paper does not link these observations to classification outcomes or to failure cases. A reviewer would expect at least one analysis of wrong predictions where the video backbone is correct and the EEG branch is not, and the other way around, to justify the structured fusion. Finally, the paper is long and dense and some implementation choices appear late in the appendix. A more compact main text would help readers who want to re implement the method quickly.

**Questions:**

1-The model forms the second stage graph with two global nodes and K region nodes and then directly projects every node to the label space. Why was this direct projection chosen over first producing a unified feature and then applying a task head.
2-It would be helpful to know whether the gains come from graph reasoning or from letting regions vote directly for the label. The correlation alignment is applied to batches of unimodal embeddings. What is the effect of the batch size on this loss and did the authors try instance wise alignment or moving average statistics to make it less sensitive to the composition of a batch.
3-In the Emognition dataset there are only four EEG channels. In this case the first stage graph is almost trivial. Can the authors report how much of the improvement on Emognition comes from the alignment loss alone.
4-In the benchmark comparison the existing multimodal models are audio and video systems that were re purposed because audio is structurally similar to EEG. Have the authors tried to re implement these models with the same EEG spectrogram encoder used in EVER in order to separate architectural gains from encoder gains. For subject independent evaluation it is important to consider domain shift across subjects. Did the authors observe subject specific clustering of the embeddings before and after alignment.
5-Figure 2 shows a nicer shared space, yet a quantitative measure such as class conditional Fréchet distance across subjects would make the claim stronger.

---

> ### Author Response · Authors · 2025-11-24
> **Response to Reviewer yFBv 's Weaknesses**
>
> We truly appreciate that Reviewer-yFBv acknowledges the practical significance and careful design of EVER, particularly highlighting the hierarchical graph construction, structured EEG–video fusion, and the thorough experimental evaluation with extensive benchmark. Please kindly refer to the following responses addressing your concerns regarding the weaknesses.
>
> ---
>
> ### 1) Unified framework and the dependency of specific backbone
>
> We respectfully clarify that EVER is inherently backbone-agnostic. Although AudioMamba and AdaMAE were chosen as practical backbones that offer a favorable balance between accuracy and computational efficiency, the framework itself does not depend on these specific models.
>
> Furthermore, as stated in our main manuscript, our benchmarking primarily focuses on widely adopted transformer-based backbones. Thus, basic CNN encoders were not included in the initial experiments. To more comprehensively demonstrate the generality of EVER, we will additionally evaluate our framework using other backbones included in our benchmarking setup.
>
> ---
>
> ### 2) Dataset size and generalization
>
> We agree that large-scale EEG–video paired datasets would be highly valuable. However, such datasets are not yet available. Although we already surveyed the paired EEG-video datasets for emotion recognition, the available synchronized datasets were limited. Therefore, EVER covers all publicly available EEG–video paired datasets: Emognition, MDMER, and EAV. Our contribution is the first unified multi-dataset evaluation for EEG–video emotion recognition, offering a standardized and comparable setting across diverse recording conditions. Regarding large-scale unlabeled synchronized recordings and missing-frame scenarios, we view these as promising directions for future work.
>
> ---
>
> ### 3) Brain-region assignment and robustness to channel variations
>
> In our benchmark setup, Emognition dataset is collected using a wearable 4-channel Muse 2 device, which directly represents the low-channel settings commonly found in practical consumer-grade EEG systems. Our proposed BIH-GCN demonstrates consistent performance improvements on Emognition dataset despite the extremely limited channel count as shown in Tabs. 3 and 4 in our main manuscript. This indicates that our framework is already robust to sparse electrode setups.
>
> Regarding the concern about potential “errors in region assignment”, we clarify that under the 10–20 system, each EEG electrode has a fixed anatomical location, and the mapping from channels to regions is deterministic. The 10–20 system provides a standardized spatial partition covering the full brain surface. Therefore, misassignment of channels to regions does not occur. Moreover, the three datasets in our benchmark employ different device configurations (4, 18, and 30 channels), and EVER maintains consistent performance across all of them, further supporting the robustness of our design under varying channel densities.
>
> ---
>
> ### 4) Justification and alternatives for correlation-based alignment
>
> We thank the reviewer for the suggestion to compare correlation-based alignment with alternative alignment objectives. While contrastive alignment requires well-defined positive and negative pairs, EEG–video features are strictly one-to-one paired, and constructing negatives artificially may introduce unstable optimization. Similarly, MMD relies on distributional assumptions (e.g., Gaussian kernels) that do not hold for heterogeneous EEG and video embeddings and has been observed to perform suboptimally in prior work [1].
>
> In contrast, our proposed correlation alignment is stable across datasets and well-suited for aligning fundamentally different representations. The ablation experiment in Tab. 3 already demonstrates consistent performance improvements. We will additionally compare with alternative feature alignment methods (e.g., CORAL) to further validate our choice in later comment.
>
> ---
>
> ### 5) Interpretation analysis and failure cases
>
> In the later comment, we will include several failure-case illustrating situations where: 1) the video modality predicts correctly while the EEG modality does not, 2) the EEG modality predicts correctly while the video modality does not. These examples will further highlight the importance of structured fusion integrating complementary information from both modalities.
>
> ---
>
> ### 6) Length of the paper and late implementation details
>
> We acknowledge this concern. In the camera-ready version, we will reorganize and move essential implementation details from the appendix to the main text for improved readability.
>
> ---
> * [1] Hongliang Yan et al., Mind the class weight bias: Weighted maximum mean discrepancy for unsupervised domain adaptation. CVPR, 2017.

---

> ### Author Response · Authors · 2025-11-24
> **Response to Reviewer yFBv 's Questions**
>
> Please kindly refer to the following responses addressing your concerns regarding the questions.
>
> ---
>
> ### 1) Direct projection vs. unified feature with additional task head
>
> In typical GCN architectures, each node already undergoes graph convolution operations, which aggregate and transform features from neighboring nodes. Because these graph convolutions inherently perform feature learning, an additional task-specific projection or separate unified feature aggregation is not strictly necessary. Directly projecting each node to the label space is a standard and effective practice in many GCN-based methods [1, 2, 3], allowing the network to leverage the learned node-wise representations without introducing redundant transformations.
>
> Introducing an extra unified feature could potentially dilute the information captured by individual nodes and reduce interpretability. Therefore, our design choice is both theoretically justified and empirically effective, as evidenced by the consistent improvements reported in our experiments.
>
> ---
>
> ### 2-1) Gains from graph vs. regional voting
>
> The observed improvement arises from the combination of graph reasoning and region-level contributions. As shown in Tab. 4 in our main manuscript, removing the each step of our BIH-GCN leads to a clear performance degradation, confirming the importance of both components.
>
> ### 2-2) Batch size sensitivity of correlation alignment
>
> Regarding batch size, our correlation alignment is formulated using batch-level covariance [4, 5], which is the standard and most stable approach in the cross-modal alignment literature. While moving-average covariance or instance-wise alignment could in principle be used, moving-average statistics tend to suppress batch-level variability and can lead to representation collapse, and instance-wise alignment often produces unstable gradients in heterogeneous multi-modal settings. Therefore, we adopt the batch-level formulation as it provides both stability and reliable alignment across datasets.
>
> Moreover, in the case of EEG + video integration, we empirically observed that training becomes highly unstable under small batch sizes. We tested batch sizes of 4, 6, and 8 across all three datasets, and both our model and other benchmark networks frequently collapsed to predicting a single class or failed to converge at all. Therefore, ensuring global training stability is substantially more critical than minimizing the sensitivity of alignment loss to specific batch compositions.
>
> ---
>
> ### 3) Emognition dataset and trivial first-stage graph
>
> Although Emognition dataset contains only 4 EEG channels, the proposed BIH-GCN still provides measurable improvements. As already reported in Tab. 3, the proposed correlation alignment alone contributes a substantial portion of the performance gain. The additional effect of the first-stage graph is systematically analyzed in Tab. 4, demonstrating that even with a minimal number of channels, hierarchical graph reasoning further enhances performance beyond what alignment alone achieves.
>
> ---
>
> ### 4-1) Encoder effects vs. architecture effects
>
> To ensure fairness, we apply the same STFT processing across all EEG backbones for the benchmark including audio-video networks, such as TVLT and HicMAE. This allows us to isolate architectural improvements from backbone-specific gains, and we will clarify this point in the revised version.
>
> ### 4-2) Subject-wise domain shift
>
> While domain shift across subjects would be an important consideration, in our case the absolute performance of all emotion recognition networks are relatively low, and the public available datasets contain a limited number of subjects. Moreover, the train/test splits are already subject-independent. As a result, subject-specific clustering of embeddings is not a primary factor affecting the evaluation.
>
> ---
>
> ### 5) Need for quantitative shared-space metrics
>
> We would like to clarify that Fig. 2 in our main manuscript does not depict differences across subjects; rather, it illustrates the gap between EEG and video features. This figure provides qualitative evidence that the alignment brings embeddings from the two modalities into a coherent shared space. Our primary objective in this work is subject-independent emotion recognition, not cross-subject comparison. Therefore, metrics such as class-conditional Fréchet distance across subjects are not central to our study.
>
> ---
>
> * [1] Thomas N Kipf and Max Welling. Semi-supervised classification with graph convolutional networks. ICLR, 2017.
> * [2] Zhao-Min Chen et al., Multi-label image recognition with graph convolutional networks. CVPR, 2019.
> * [3] Matthew Korban and Xin Li. Ddgcn: A dynamic directed graph convolutional network for action recognition. ECCV, 2020.
> * [4] Baochen Sun and Kate Saenko. Deep coral: Correlation alignment for deep domain adaptation. ECCV, 2016.
> * [5] Ya Jing et al., Cross-modal cross-domain moment alignment network for person search. CVPR, 2020.

---

> ### Author Response · Authors · 2025-12-01
> **Additional Experiments and Analyses**
>
> We conducted extensive experiments and analyses to validate the backbone-agnostic nature of our framework, assess the effectiveness of the proposed correlation-based alignment, and examine failure cases. Please refer to the details below.
>
> ---
>
> ### Weakness 1) Our framework with other backbones
>
> To further verify that EVER is inherently backbone-agnostic, we conducted additional experiments by replacing the AdaMAE and AudioMamba backbones used in the main paper with three alternative architectures. As shown in Table, proposed architecture consistently yields substantial performance gains regardless of the backbone choice. Notably, combining different backbone pairs (e.g., ViViT + Crossformer, VideoMAE + Crossformer) with our proposed BIH-GCN and correlation-based alignment results in clear improvements across all evaluation metrics. These findings confirm that EVER does not rely on any specific architectural property of a particular backbone and is broadly applicable to a wide range of backbones.
>
> | Method                 | Acc (Emognition) | UAR (Emognition) | W-F1 (Emognition) | Acc (EAV) | W-F1 (EAV) |
> |------------------------|----------------|----------------|------------------|---------|-----------|
> | ViViT                  | 0.262          | 0.244          | 0.244           | 0.395   | 0.300     |
> | VideoMAE               | 0.261          | 0.224          | 0.229           | 0.408   | 0.446     |
> | Crossformer            | 0.294          | 0.246          | 0.234           | 0.358   | 0.345     |
> | ViViT + Crossformer + Ours    | 0.318          | 0.263          | 0.344           | 0.455   | 0.416     |
> | VideoMAE + Crossformer + Ours | 0.307          | 0.258          | 0.357           | 0.459   | 0.429     |
>
> ---
>
> ### Weakness 4) Justification of feature alignment with other methods
>
> To evaluate the effectiveness of our proposed feature alignment loss, we replaced $L_{align}$ with two alternatives (i.e., MMD and CORAL), and conducted experiments on the Emognition dataset. The results show that both substitutions lead to clear performance degradation across all metrics. These findings indicate that proposed correlation-based alignment approach is effective for handling the heterogeneity between the two modalities.
>
> | Method                 | Acc            | UAR            | W-F1           |
> |------------------------|----------------|----------------|----------------|
> | MMD                    | 0.326   | 0.319          | 0.306          |
> | CORAL                  | 0.318          | 0.321    | 0.308    |
> | Ours ($L_{align}$)     | **0.335**      | **0.359**      | **0.355**      |
>
> **Bold = Best**
>
> ---
>
> ### Weakness 5) Analysis of failure cases
> We observed situations where the video modality makes correct predictions while the EEG modality fails, and vice versa.
>
> EEG succeeds while the video modality fails in scenarios involving subtle, transient emotional responses such as brief internalized reactions or micro-level arousal changes, which are not clearly reflected in facial expressions or occur between video frames. Similarly, when participants maintain a neutral facial appearance despite strong internal reactions, the video modality tends to misclassify emotions, whereas EEG can still capture the underlying cognitive–affective dynamics.
>
> Conversely, the video modality succeeds while EEG fails in cases where the user exhibits overt, behaviorally clear emotional expressions, but EEG signals become unreliable due to noise, motion artifacts, or poor electrode contact. In these moments, facial expressions provide a stable cue, while the EEG modality becomes ambiguous or inconsistent.
>
> These contrasting patterns reinforce the necessity of a structured fusion approach that effectively integrates the complementary strengths of EEG and video. The failure-case analysis highlights that neither modality is sufficient alone, and robust emotion recognition emerges only when the two information sources are combined.

---

### Author Response · Authors · 2025-12-03
**Revision Uploaded**

We have updated the revised version of the main manuscript. Newly added contents and tables are highlighted in red for your convenience. Most of the additions correspond to the extra experiments requested by the reviewers.

We hope these revisions adequately address the reviewers’ comments. For more detailed rebuttal responses and clarifications, please kindly refer to the comments corresponding to each reviewer.

---

### Meta-Review · Area_Chair_1CFR · 2025-12-27

**Summary:**

This paper was reviewed by 4 experts and received 4, 4, 2, 6 as the initial ratings. The reviewers agreed that the paper introduces a novel framework that explicitly unifies video and EEG representations, the proposed model architecture is carefully designed, the experimental results are promising and the ideas in the paper are clearly expressed.

Reviewer 95fJ raised a concern about the novelty of the proposed method. While the authors have clarified this in their response, the AC feels that a more thorough explanation is necessary to understand the novelty. It is not very clear how Stage 1 of the proposed BIH-GCN differs from other existing methods; moreover, a detailed analysis of the superiority of Stage 2 compared to other GCNs that also consider global and local connections (such as the references mentioned in the review) is necessary to convincingly establish the novelty of the proposed method.

Reviewer 8y7d mentioned that the paper does not provide a detailed analysis of the balance between the performance and computational complexity of the proposed method. In their response, the authors have explained that their model attains good accuracy with a moderate number of parameters (referring to Table 2 in the paper). However, the AC feels that a more thorough comparative analysis of the computational complexity (in terms of the FLOP counts) is necessary to appropriately understand the usefulness of the proposed framework. It is a bit difficult to assess the performance vs. computational complexity trade-off from the presented results.

Reviewer wXbG raised a concern that the model may get biased toward identity-related facial attributes, as the video branch employs AdaMAE pretrained on VoxCeleb. This is an important concern, but was not addressed by the authors in their response.

We appreciate the authors' efforts in meticulously responding to each reviewer’s comments and conducting the additional experiments to answer some of the reviewers' questions (such as the experiment with different backbone architectures, and the ablation studies with different feature alignment methods). However, in light of the above discussions, we conclude that the paper may not be ready for an ICLR publication in its current form. While the paper clearly has merit, the decision is not to recommend acceptance. The authors are encouraged to consider the reviewers' comments when revising the paper for submission elsewhere.

**Reviewer Concerns:**

Please see my comments above.

**Reviewer Scores:**

Reviewer yFBv -> 5

Reviewer 8y7d -> 4

Reviewer 95fJ -> 3

Reviewer wXbG -> 6

---

### Decision · Program_Chairs · 2026-01-26

Reject